# The AMPA receptor-associated protein Shisa7 regulates hippocampal synaptic function and contextual memory

**Leanne J M Schmitz[1,2‡], Remco V Klaassen[1‡], Marta Ruiperez-Alonso[3‡], Azra Elia Zamri[1,4], Jasper Stroeder[2,3], Priyanka Rao-Ruiz[1], Johannes C Lodder[3], Rolinka J van der Loo[1,2], Huib D Mansvelder[3†], August B Smit[1†], Sabine Spijker[1†*]**

[1]Department of Molecular and Cellular Neurobiology, Center for Neurogenomics and Cognitive Research, Amsterdam Neuroscience, VU University, Amsterdam, The Netherlands; [2]Sylics (Synaptologics BV), Amsterdam, The Netherlands; [3]Department of Integrative Neurophysiology, Center for Neurogenomics and Cognitive Research, Amsterdam Neuroscience, Vrije Universiteit Amsterdam, Amsterdam, The Netherlands; [4]Université de Bordeaux, Interdisciplinary Institute for Neuroscience, UMR 5297, Bordeaux, France

**\*For correspondence:**
s.spijker@vu.nl

[†]These authors also contributed equally to this work
[‡]These authors also contributed equally to this work

**Abstract** Glutamatergic synapses rely on AMPA receptors (AMPARs) for fast synaptic transmission and plasticity. AMPAR auxiliary proteins regulate receptor trafficking, and modulate receptor mobility and its biophysical properties. The AMPAR auxiliary protein Shisa7 (CKAMP59) has been shown to interact with AMPARs in artificial expression systems, but it is unknown whether Shisa7 has a functional role in glutamatergic synapses. We show that Shisa7 physically interacts with synaptic AMPARs in mouse hippocampus. *Shisa7* gene deletion resulted in faster AMPAR currents in CA1 synapses, without affecting its synaptic expression. *Shisa7* KO mice showed reduced initiation and maintenance of long-term potentiation of glutamatergic synapses. In line with this, *Shisa7* KO mice showed a specific deficit in contextual fear memory, both short-term and long-term after conditioning, whereas auditory fear memory and anxiety-related behavior were normal. Thus, Shisa7 is a bona-fide AMPAR modulatory protein affecting channel kinetics of AMPARs, necessary for synaptic hippocampal plasticity, and memory recall.
DOI: https://doi.org/10.7554/eLife.24192.001

## Introduction

In the adult brain, fast excitatory synaptic transmission is largely mediated by AMPA-type glutamate receptors (AMPARs). Activity-dependent changes in the efficacy of glutamatergic transmission depends on both pre- and postsynaptic mechanisms (*Fioravante and Regehr, 2011*; *Castillo, 2012*; *Huganir and Nicoll, 2013*). In the post-synapse this process is mainly determined by regulation of the number and biophysical properties of synaptic AMPARs (*Conti and Weinberg, 1999*; *Jonas, 2000*; *Carroll et al., 2001*; *Bredt and Nicoll, 2003*; *Choquet and Triller, 2003*; *Shepherd and Huganir, 2007*; *Choquet, 2010*; *Fioravante and Regehr, 2011*; *Castillo, 2012*). Activity-dependent plasticity underlying learning, memory, and synapse turnover (*Malenka and Nicoll, 1999*; *Malinow and Malenka, 2002*; *Derkach et al., 2007*; *Newpher and Ehlers, 2008*), has been shown to rely on AMPAR post-translational modifications and specific protein interactions. Over the last years, a range of mostly transmembrane proteins has been identified as components of native brain-derived AMPAR complexes (*Schwenk et al., 2012*). The function of most of these direct AMPAR interactors is hitherto unknown, however, several have been identified as AMPAR auxiliary subunits, affecting trafficking, channel kinetics, surface mobility, and activity-dependent

regulation of these receptors (*Jackson and Nicoll, 2011*; *Straub and Tomita, 2012*; *Sumioka, 2013*; *Martenson and Tomita, 2015*). The transmembrane AMPAR regulatory proteins (TARPs) (*Tomita et al., 2003*; *Rouach et al., 2005*), the Cornichon homologs (CNIH-2 and CNIH3) (*Schwenk et al., 2009*), Germ Cell-Specific Gene 1-Like (GSG1L [*Shanks et al., 2012*; *Gu et al., 2016*]), SynDIG1 (*Kalashnikova et al., 2010*), Porcn (*Erlenhardt et al., 2016*), and the recently identified members of the Shisa family, Shisa9/CKAMP44 (*von Engelhardt et al., 2010*; *Pei and Grishin, 2012*; *Karataeva et al., 2014*; *Farrow et al., 2015*) and Shisa6/CKAMP52 (*Klaassen et al., 2016*) all modulate AMPAR function in a unique manner. TARPs and cornichons decrease deactivation and desensitization rates of the activated AMPAR, and promote synaptic targeting. In contrast, GSG1L increases AMPAR deactivation and desensitization when overexpressed in HEK293 cells (*Shanks et al., 2012*) and in hippocampal CA1 pyramidal cells of *GSG1L* KO mice (*Gu et al., 2016*). This overexpression also reduces synaptic AMPAR expression and transmission (*Gu et al., 2016*).

The Shisa proteins display a distinct profile of modulation of AMPAR expression, channel kinetics and activity-dependent regulation. Shisa9 and Shisa6 similarly slow AMPAR deactivation ex vivo in the dentate gyrus and CA1 (*von Engelhardt et al., 2010*; *Khodosevich et al., 2014*; *Klaassen et al., 2016*), and both proteins slow the rate of recovery from desensitization in a heterologous expression system (*Farrow et al., 2015*; *Klaassen et al., 2016*). In contrast, Shisa9 enhances the desensitization rate and reduces short-term facilitation of glutamatergic synaptic transmission (*von Engelhardt et al., 2010*), whereas Shisa6 prevents AMPAR desensitization, thereby reducing short-term synaptic depression (*Klaassen et al., 2016*). Recently, Shisa7/CKAMP59 was identified as AMPAR interacting protein by co-immunoprecipitation with GluA1 and GluA2 in HEK293 cells (*Farrow et al., 2015*). Surprisingly, Shisa7 did not exert effect on channel kinetics, unlike Shisa6 and Shisa9. Whether Shisa7 has a functional role in glutamatergic synapses and behavior that depends on glutamatergic synaptic transmission is not known. Here, we find that Shisa7 displays strong synaptic enrichment and colocalizes with the AMPAR. We reveal that Shisa7 directly associates with AMPARs within native hippocampal protein complexes, independent of AMPAR subunit composition. By creating transgenic animals lacking the Shisa7 protein, we find that Shisa7, unlike shown for Shisa9, is important for hippocampal glutamatergic synaptic plasticity as well as contextual memory.

## Results

Shisa7 shares high structural similarity with the established AMPAR-associated proteins Shisa6, and Shisa9 (*Figure 1a*). Real-time PCR showed abundant expression of the *Shisa7* gene in the mouse brain, where it is expressed with a profile similar to AMPAR genes *Gria1* and *Gria2* during postnatal development (*Figure 1—figure supplement 1*). In situ hybridization analysis (Allen Brain Atlas; *Figure 1—figure supplement 1*) revealed widespread expression in the whole brain, except for the cerebellum, with high expression in cortex, striatum, amygdala and hippocampus CA1-3 and dentate gyrus, as was shown previously (*Farrow et al., 2015*). A *Shisa7* knockout (KO) mouse was generated, which confirmed the specificity of the Shisa7 antibody (*Figure 1—figure supplement 2*). Immunoblotting with this antibody showed highly enriched expression of Shisa7 in the cortex, as well as expression in striatum and hippocampus (*Figure 1b*). The Shisa7 protein in the hippocampus is glycosylated (*Figure 1—figure supplement 2*) leading to an increase in observed molecular weight, similar to Shisa6 (*Klaassen et al., 2016*).

The subcellular distribution of surface Shisa7 was explored by immunofluorescence staining of lentivirus-expressed Flag-Shisa7 in *Shisa7* KO primary hippocampal neurons (DIV18). Tagging was required due to insufficient specificity of the Shisa7 antibody in immunohistochemical staining. A qualitative analysis of Shisa7 localization showed a moderately high level of granular staining for Shisa7 present along the dendrites, co-localizing with a similar granular staining for the synaptic protein Homer1 (*Figure 1c*). A quantitative analysis was not possible, as we observed considerable spine loss after overexpression of Shisa7 in *Shisa7* KO and WT alike. In agreement with the co-localization of Shisa7 and Homer1, subcellular fractionation of hippocampal proteins revealed that Shisa7 was highly enriched in the Triton-X100-insoluble postsynaptic density (PSD) fraction, in which it co-purified with PSD-95, GluA2 and GluN2A (*Figure 1d*).

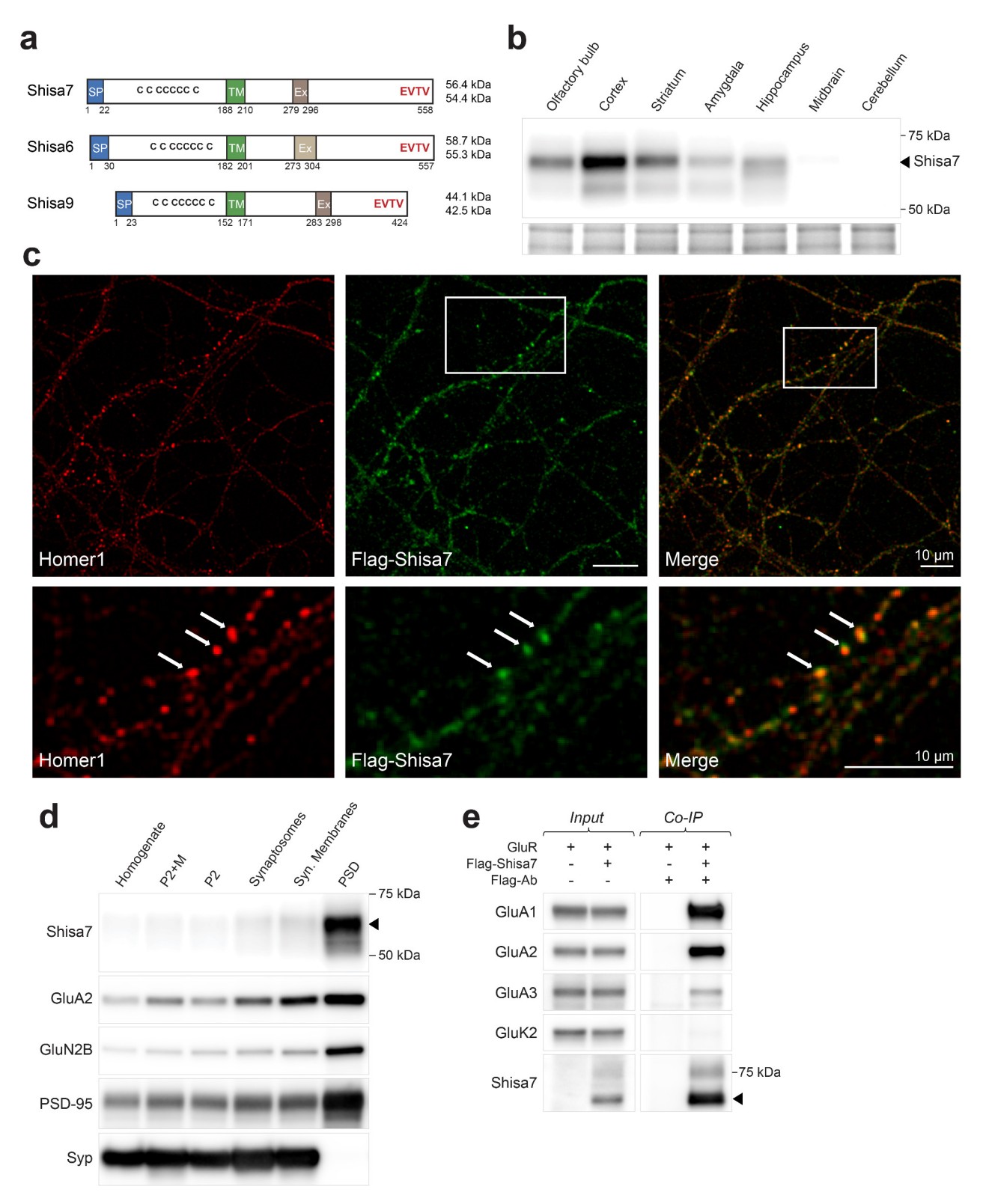

**Figure 1.** Shisa7 is a type-I transmembrane protein interacting with AMPA-type receptors. (a) Shisa7 is closely related to the AMPAR auxiliary subunit Shisa6 and Shisa9, bearing a signal peptide (SP; 22 amino acids), an extracellular domain with conserved cysteine-rich motif, a single transmembrane region (TM), and an intracellular domain with a type II PDZ-ligand motif (EVTV). Exon4 (Ex) is an alternative-splice region in *Shisa7* and *Shisa9*, whereas this is Exon3 in *Shisa6*. The predicted molecular weight of the two mature Shisa7 protein variants is ~56 and ~54 kDa, and that of Shisa6 ~59 and ~55

*Figure 1 continued on next page*

*Figure 1 continued*

kDa (*Klaassen et al., 2016*). (b) Shisa7 is highly enriched in the cortex, olfactory bulb and hippocampus, and absent in cerebellum, as measured in crude synaptic membrane fractions. Despite the presence of both transcript variants (see *Figure 1—figure supplement 1*), the indicated protein band (~68 kDa; arrow head) is dominant in the hippocampus. Lower panel depicts the loading control, that is, total crude synaptic membrane protein. (c) Immunohistochemistry of primary *Shisa7* KO hippocampal neurons (DIV14) after viral overexpression of Flag-Shisa7 shows Shisa7 expression (green) in endogenous Homer1-positive puncta (red). The lower panels show a zoom-in (white box in upper panel). The overlay of the two channels is shown, scale bars are indicated. (d) Biochemical fractionation (homogenate (H), crude synaptic membranes (P2; with and without microsomes (M)), synaptosomes (SS), synaptic membranes (SM) and postsynaptic density fraction (PSD; Triton X-100 insoluble fraction) of mature mouse hippocampus reveals an enrichment of Shisa7 in the PSD together with GluA2, GluN2B, PSD-95, and this pattern is distinct from the presynaptic marker Synaptophysin (Syp). (e) Precipitation of Flag-Shisa7 (~60 kDa) from HEK293 cells using a Flag antibody shows that upon co-expression it binds directly to homomeric GluA1, GluA2 and GluA3 receptors, whereas having minimal affinity for GluK2. For complete blots, in addition to those with higher exposure, see *Figure 1—figure supplement 3*. The input controls represent 2% of the total lysate.

DOI: https://doi.org/10.7554/eLife.24192.002

The following source data and figure supplements are available for figure 1:

**Source data 1.** Sequence of DNA primers.
DOI: https://doi.org/10.7554/eLife.24192.006
**Figure supplement 1.** *Shisa7* gene expression.
DOI: https://doi.org/10.7554/eLife.24192.003
**Figure supplement 2.** Generation of *Shisa7* KO mice and antibody testing.
DOI: https://doi.org/10.7554/eLife.24192.004
**Figure supplement 3.** Whole immunoblot compilation.
DOI: https://doi.org/10.7554/eLife.24192.005

## Shisa7 interacts with AMPARs and directly binds PSD-95

Shisa7 is an AMPAR-interacting protein in HEK293 cells without subunit specificity (*Figure 1e*). Next, we addressed whether Shisa7 is an AMPAR-interacting protein in the hippocampus. First, the presence of AMPAR subunits in native hippocampal Shisa7 protein-complexes was determined by immunoprecipitation from the DDM-extracted crude synaptic membrane fraction followed by mass spectrometry. Under these conditions, Shisa7 formed a stable association with AMPAR subunits GluA1, GluA2, and GluA3 (*Table 1*, *Table 1—source data 1*). Based on the established AMPAR interactome (*Schwenk et al., 2012*; *Chen et al., 2014*), we furthermore identified Shisa6 in these native Shisa7 complexes.

Mass spectrometry of native hippocampal Shisa7 complexes identified PSD-95 as the most prominent PDZ domain-containing interactor (*Table 1*), and the direct nature of this interaction was confirmed by direct two-hybrid assay (*Figure 1—figure supplement 2*). Finally, we established that the association between Shisa7 and PSD-95 is dependent on the Shisa7 type II PDZ-ligand motif (EVTV; *Figure 1—figure supplement 2*), similar as was reported for Shisa9 (*Karataeva et al., 2014*) and Shisa6 (*Klaassen et al., 2016*).

## Shisa7 affects AMPAR recovery from desensitization in HEK293 cells

Farrow et al. (*Farrow et al., 2015*) recently reported that despite direct interactions between Shisa7 (CKAMP59) and AMPAR subunits in HEK293 cells, Shisa7 had no effect on the biophysical properties of AMPARs. Similar to that study, we can confirm here that in our hands Shisa7 did not affect most properties of fast glutamate application (1 ms; 1 mM) on AMPARs in HEK293 cells. Co-expression of Shisa7 did not alter AMPAR deactivation, rise time, or rectification (*Figure 2—figure supplement 1*), in contrast to Shisa6 (*Klaassen et al., 2016*) and Shisa9 (*von Engelhardt et al., 2010*). Yet, Shisa7 had a significant effect in lowering desensitization properties upon prolonged glutamate application (1 s, 1 mM; *Figure 2a,b*) of homomeric AMPARs, i.e., on the desensitization time (t-test; p=0.007) and steady state current (t-test; p=0.012). In addition, Shisa7 affected the recovery from desensitization using two consecutive 1 ms glutamate (1 mM) applications with variable time interval (*Figure 2c,d*). Shisa7 slowed down recovery from desensitization, resulting in an increase in the time constant of recovery($\tau_{recovery}$: 82.6 ± 6.7 ms (GluA1) *vs.* 119.1 ± 10.9 ms (GluA1 +Shisa7); t-test; p=0.006; *Figure 2d*).

**Table 1.** Shisa7 is a native interactor of hippocampal AMPARs.

Shisa7 complexes were immunoprecipitated from the hippocampi of *Shisa7* WT and KO animals (DDM-extracted crude synaptic membranes; n = 3 IPs per genotype) and subjected to mass spectrometric analysis. The following protein categories were indicated: Shisa7, AMPAR subunits, established non-PDZ AMPAR-interacting proteins (***Schwenk et al., 2012***), PDZ domain-containing interactors. Percent coverage, percent of the database protein sequence covered by all matching peptides. Values of statistical significance upon performing a Student's t-test with permutation-based FDR analysis (S0 = 1, FDR < 0.05 (bold)) are indicated. Note the identification of a peptide specific for exon 4 containing Shisa7 (NLYNTMKPSNLDwNLHYNVNSPK; diamond indicates exon3-exon4 boundary). The full list of identified protein groups is reported in ***Table 1—source data 1***, and the statistical analysis is provided in ***Table 1—source data 2***, which includes a volcano-plot of differentially IP-ed proteins.

| Gene name | Uniprot recommended protein name(s) | Uniprot ID | PDZ-domains | Number of unique peptides | | | | | | LFQ intensity | | | | | | Average KO LFQ intensity | Average WT LFQ intensity | Average KO/WT LFQ intensity | Average WT/KO LFQ intensity | T-test significant FDR 0.01 = ++ DR 0.05 = + | T-test q-value | Percent coverage |
|---|---|---|---|---|---|---|---|---|---|---|---|---|---|---|---|---|---|---|---|---|---|---|
| | | | | KO1 | KO2 | KO3 | WT1 | WT2 | WT3 | KO1 | KO2 | KO3 | WT1 | WT2 | WT3 | | | | | | | |
| Shisa7 | Protein Shisa-7 | Q8C3Q5 | 0 | 1 | 1 | 2 | 17 | 18 | 16 | 865 | 873 | 1750 | 319820 | 305190 | 346730 | 1163 | 323913 | 0.4% | 278.58 | ++ | **0.0000** | 39.2 |
| | Protein Shisa-7: Exon4-specific peptide NLYNTMKPSN LDNLHYNVNSPK | - | - | 0 | 0 | 0 | 1 | 0 | 0 | - | - | - | - | - | - | - | - | - | - | - | - | - |
| Gria1 | Glutamate receptor 1 | P23818 | 0 | 0 | 0 | 0 | 10 | 12 | 12 | 5921 | 0 | 3217 | 17828 | 52304 | 54212 | 4569 | 41448 | 11.0% | 9.07 | ++ | **0.0000** | 22.8 |
| Gria2 | Glutamate receptor 2 | P23819 | 0 | 2 | 1 | 0 | 17 | 17 | 17 | 2743 | 0 | 0 | 104860 | 101430 | 93097 | 2743 | 99796 | 2.7% | 36.39 | ++ | **0.0000** | 36.4 |
| Gria3 | Glutamate receptor 3 | Q9Z2W9 | 0 | 0 | 0 | 0 | 4 | 4 | 5 | 0 | 0 | 0 | 6529 | 7068 | 10102 | 0 | 7900 | 0.0% | NaN | ++ | **0.0000** | 17.7 |
| Cacng8 | Voltage-dependent calcium channel gamma-8 subunit; TARP gamma-8 | Q8VHW2 | 0 | 2 | 2 | 1 | 2 | 1 | 5 | 4023 | 0 | 0 | 0 | 0 | 3962 | 4023 | 3962 | 101.5% | 0.98 | - | N/A | 30.5 |
| Olfm1 | Noelin | O88998 | 0 | 1 | 0 | 0 | 0 | 2 | 2 | 0 | 1076 | 0 | 0 | 1004 | 779 | 1076 | 892 | 120.6% | 0.83 | - | N/A | 7.0 |
| Prrt1 | Proline-rich transmembrane protein 1; SynDIG4 | O35449 | 0 | 0 | 0 | 0 | 1 | 1 | 0 | 0 | 0 | 0 | 969 | 0 | 0 | 0 | 969 | 0.0% | NaN | - | N/A | 7.2 |
| Prrt2 | Proline-rich transmembrane protein 2 | E9PUL5 | 0 | 1 | 0 | 1 | 1 | 1 | 1 | 0 | 333 | 0 | 318 | 0 | 202 | 333 | 260 | 127.9% | 0.78 | - | N/A | 3.8 |
| Rap2b | Ras-related protein Rap-2b | P61226 | 0 | 1 | 1 | 1 | 0 | 0 | 0 | 0 | 0 | 2588 | 0 | 0 | 0 | 2588 | 0 | NaN | 0.00 | - | N/A | 12.0 |
| Shisa6 | Protein shisa-6 homolog | Q3UH99 | 0 | 0 | 0 | 3 | 1 | 3 | 1 | 0 | 0 | 0 | 2666 | 3580 | 2136 | 0 | 2794 | 0.0% | NaN | + | 0.0306 | 9.9 |

*Table 1 continued on next page*

*Table 1 continued*

| Gene name | Uniprot recommended protein name(s) | Uniprot ID | PDZ-domains | Number of unique peptides | | | | | | LFQ intensity | | | | | | Average KO LFQ intensity | Average WT LFQ intensity | Average KO/WT LFQ intensity | Average WT/KO LFQ intensity | T-test significant FDR 0.01 = ++ 0.05 = + | T-test q-value | Percent coverage |
|---|---|---|---|---|---|---|---|---|---|---|---|---|---|---|---|---|---|---|---|---|---|---|
| | | | | KO1 | KO2 | KO3 | WT1 | WT2 | WT3 | KO1 | KO2 | KO3 | WT1 | WT2 | WT3 | | | | | | | |
| Dlg1 | Disks large homolog 1; SAP97 | Q811D0 | 3 | 0 | 1 | 1 | 3 | 0 | 0 | 0 | 0 | 0 | 2126 | 0 | 0 | 0 | 2126 | 0.0% | NaN | - | N/A | 4.4 |
| Dlg3 | Disks large homolog 3; SAP102 | P70175 | 3 | 0 | 0 | 0 | 0 | 2 | 2 | 0 | 0 | 0 | 0 | 2441 | 0 | 0 | 2441 | 0.0% | NaN | - | N/A | 4.8 |
| Dlg4 | Disks large homolog 4; PSD95 | Q62108 | 3 | 0 | 0 | 0 | 9 | 8 | 7 | 0 | 0 | 0 | 11835 | 13410 | 9284 | 0 | 11510 | 0.0% | NaN | ++ | 0.0000 | 21.7 |
| Magi2 | Membrane-associated guanylate kinase, WW and PDZ domain-containing protein 2 | Q9WVQ1 | 6 | 0 | 0 | 0 | 1 | 1 | 0 | 0 | 0 | 0 | 1558 | 1558 | 0 | 0 | 1558 | 0.0% | NaN | - | N/A | 1.6 |

DOI: https://doi.org/10.7554/eLife.24192.007

The following source data available for Table 1:

**Source data 1.** Maxquant analysis of hippocampal Shisa7 immunoprecipitation experiments, as detailed in the Materials and methods section.
The Maxquant 'proteinGroups.txt' output file was supplemented with 'LFQ intensity _ Average KO', 'LFQ intensity _ Average WT', and the 'KO/WT' and 'WT/KO' ratios thereof.
DOI: https://doi.org/10.7554/eLife.24192.008

**Source data 2.** Statistical analysis of hippocampal Shisa7 immunoprecipitation data, as detailed in the Materials and methods section.
Tab 1: In summary, the Maxquant 'proteinGroups.txt' output file was imported into Perseus, and processed in the following manner: (1) Removal of ' Reverse', 'Potential contaminant', and 'Only identified by site' protein groups; (2) Log(2) transformation of all LFQ intensity values; (3) Removal of protein groups with less than three valid 'Log(2) LFQ intensity' values in either the WT or KO groups; (4) Imputation of missing values (8,6% of the population) from a normal distribution (width 0.3, down shift 1.8, whole matrix); (5) Performing a Student's t-test followed by permutation-based FDR analysis (S0 = 1, FDR = 0.01, 2500 permutations). Tab 2: Visualization of the data by means of Histogram and Vulcanoplot is presented in the additional sheets. Tab 3: Distribution of LFQ intensities after replacing missing values from a normal distribution. The imputed value distribution is depicted in red.
DOI: https://doi.org/10.7554/eLife.24192.009

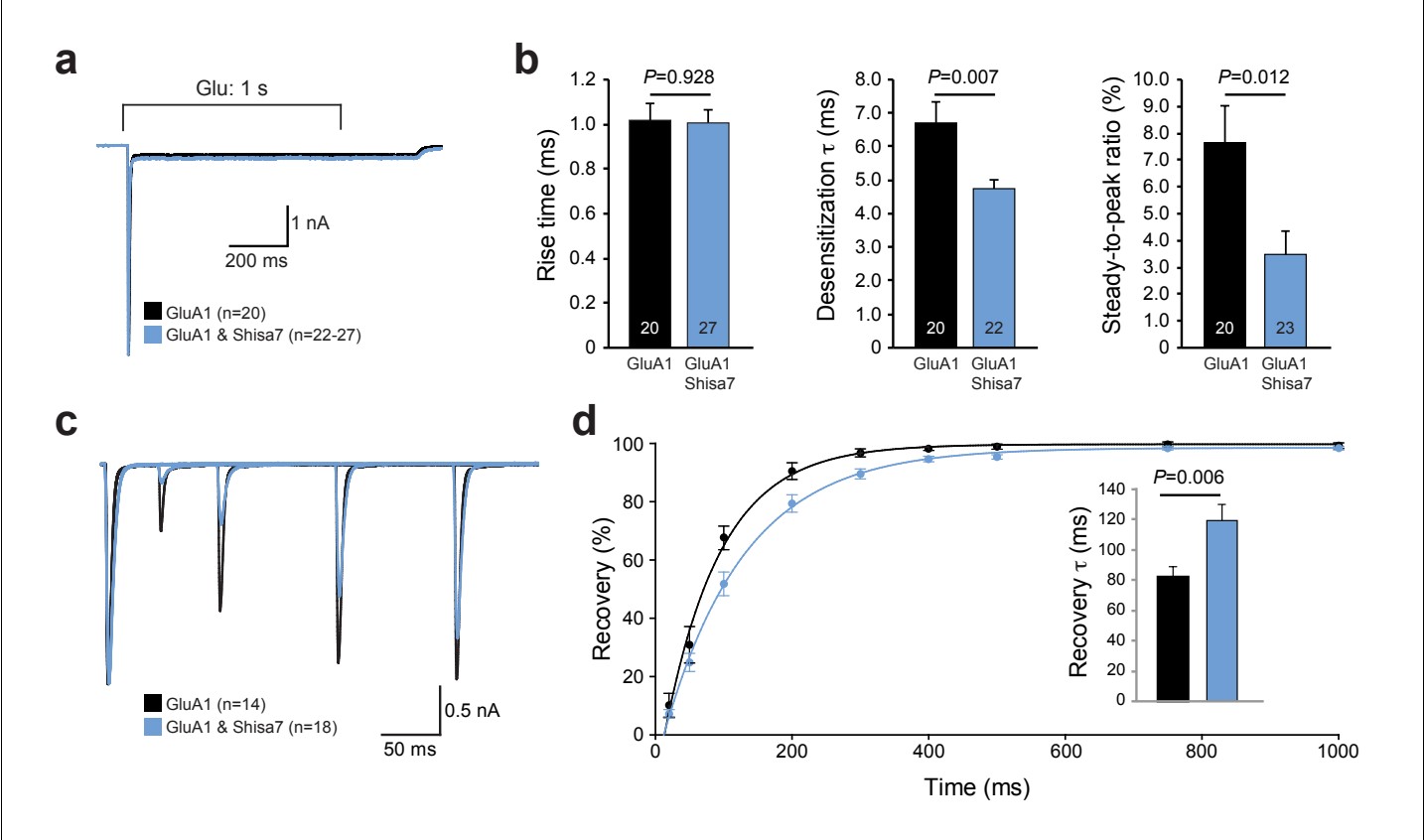

**Figure 2.** Shisa7 effects on AMPAR desensitization rate and recovery from desensitization. (**a**) Peak-scaled example trace of whole-cell recordings from HEK293 cells expressing a homomeric GluA1-containing AMPAR channel in the absence (black) or presence (blue) of Shisa7. Currents were evoked by direct application of 1 mM glutamate during 1 s. (**b**) Bar graphs (mean ±SEM) summarize changes in rise time (t45 = 0.091; p=0.928), desensitization time constant (t25.38 = 2.922; p=0.007) and steady-state AMPAR-mediated currents (t41 = 2.638; p=0.012). (**c**) Example trace of repeated 1 ms glutamate application from HEK293 cells expressing a homomeric GluA1-containing AMPAR channel in the absence (black) or presence (blue) of Shisa7. (**d**) Recovery of desensitization (two 1 ms glutamate application with inter-pulse interval of 20, 50, 100, 200, 300, 400, 500, 750, and 1000 ms) from HEK293 cells expressing a homomeric AMPAR channel in the absence (black) or presence (blue) of Shisa7. Inset shows a significant increase in $\tau_{recovery}$ in the presence of Shisa7(t23.35 = −3.022; p=0.006).

DOI: https://doi.org/10.7554/eLife.24192.010

The following figure supplement is available for figure 2:

**Figure supplement 1.** Shisa7 does not alter AMPAR kinetics in vitro or alter membrane properties in vivo.

DOI: https://doi.org/10.7554/eLife.24192.011

## Shisa7 alters synaptic AMPAR current kinetics, but not short-term plasticity

To test whether Shisa7 affects AMPAR function in the native environment of the hippocampus, we first recorded AMPAR miniature EPSCs (mEPSCs) in CA1 pyramidal cells in acute hippocampal slices of WT and *Shisa7* KO mice (*Figure 3a–d*). In WT pyramidal neurons, the decay of mEPSCs was slower than in *Shisa7* KO (7.48 ± 0.22 *vs.* 5.54 ± 0.07, unpaired t-test, p<0.001), whereas rise time, amplitude, and frequency were similar between WT and *Shisa7* KO (MWU-test, p=0.462; unpaired t-test, p=0.297; MWU-test, p=0.469, respectively). Thus, lack of Shisa7 resulted in faster deactivation kinetics of AMPARs, without affecting amplitude and frequency of AMPAR-mediated currents. In addition, basic neurophysiological parameters were not affected (series resistance 141 ± 12 *vs.* 130±13 MΩ, unpaired t-test, p=0.541 and resting membrane potential: −70.7 ± 1.2 *vs.* −71.9 ± 3.8 mV, unpaired t-test, p=0.768; *Figure 3—figure supplement 1*).

To confirm that a lack of Shisa7 did not induce changes in the abundance of membrane-localized AMPARs (or their core interactors), similar to what was observed for Shisa6 (*Klaassen et al., 2016*),

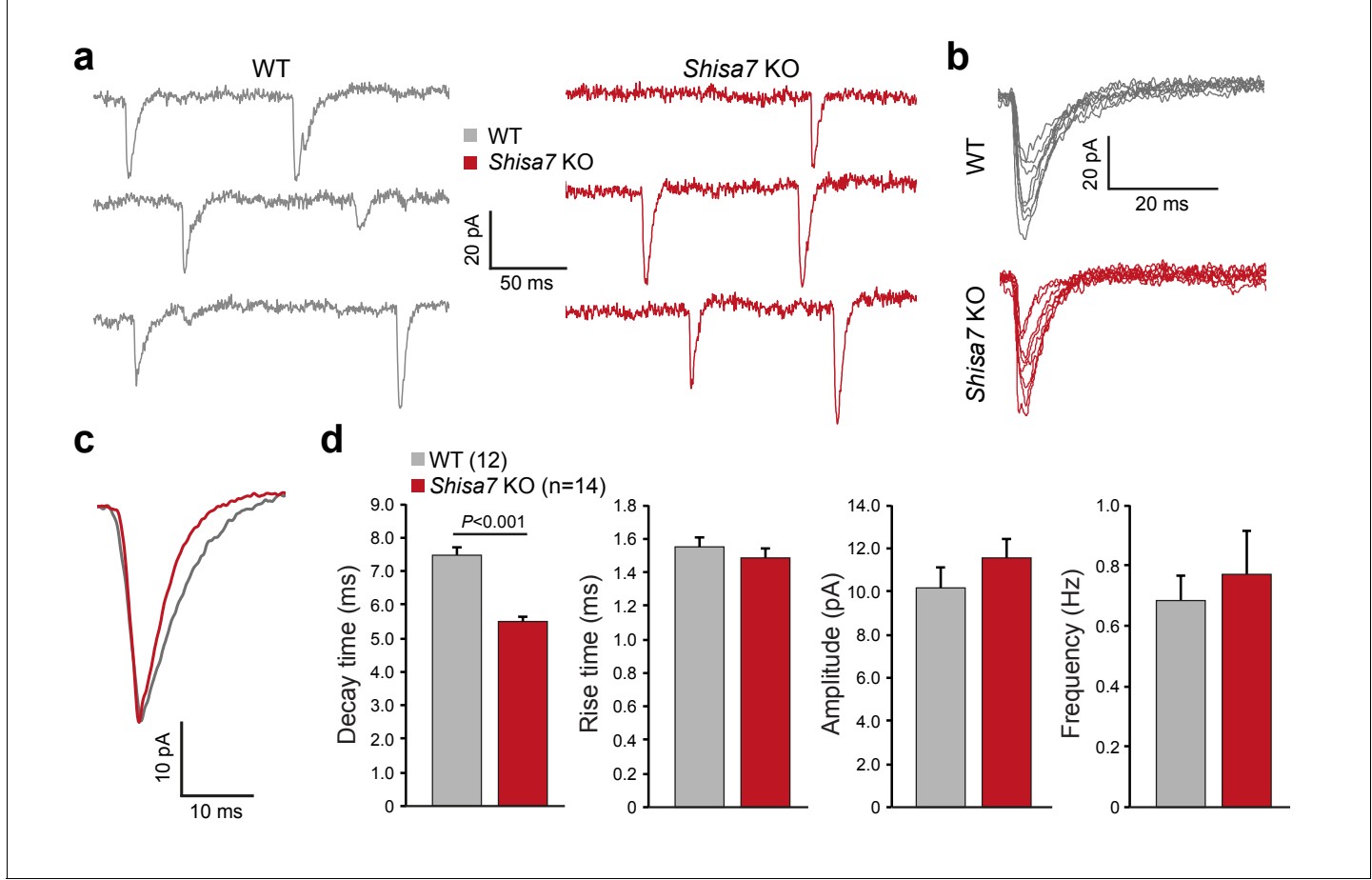

**Figure 3.** Shisa7 prolongs synaptic AMPAR currents. (**a**) Example traces of mEPSC recordings from CA1 pyramidal cells of *Shisa7* KO animals and WT littermates. (**b,c**) Superimposed spontaneous synaptic currents (**b**), and average synaptic currents (**c**) of *Shisa7* KO animals and WT littermates. (**d**) Bar graphs (mean ±SEM) of *Shisa7* KO (n = 14 cells from six animals) and WTs (n = 12 cells from six animals) show decreased decay time of mEPSCs (WT: 7.48 ± 0.22; *Shisa7* KO: 5.50 ± 0.17; p<0.001). Rise time, amplitude and frequency were not significantly affected (p=0.462, p=0.165, p=0.992, respectively).

DOI: https://doi.org/10.7554/eLife.24192.012

The following figure supplement is available for figure 3:

**Figure supplement 1.** Shisa7 does not alter membrane properties in vivo.

DOI: https://doi.org/10.7554/eLife.24192.013

immunoblotting of the hippocampal synaptic membrane fraction from *Shisa7* WT and KO mice was performed. This revealed no difference in the levels of (subunits of) the AMPAR, NMDAR, PSD-95, TARPs, CNIH, Shisa9 or Shisa6 present at the synapse (*Figure 4a*, *Figure 4—figure supplement 1*). Also, the expression of synaptic GluA2, as defined by punctate expression of Homer1, in the absence of *Shisa7* (*Figure 4b,c*) was similar (DIV14: 1.00 ± 0.04 *vs.* 1.02 ± 0.05; DIV21: 1.00 ± 0.03 *vs.* 1.10 ± 0.10), albeit that there was between culture variation (*Figure 4—figure supplement 1*). In conclusion, the absence of Shisa7 slowed down AMPAR current deactivation, without gross molecular synaptic rearrangements.

A lack of difference in AMPAR amplitude was also observed in evoked AMPAR EPSCs that were measured by stimulating Schaffer collaterals during whole-cell recordings from CA1 pyramidal neurons, while blocking GABARs with bicuculline (10 μM) and NMDARs with AP5 (100 μM) (*Figure 5*). There was no difference in input-output curves of AMPAR eEPSCs between WT and *Shisa7* KO brain slices (p=0.126; *Figure 5—figure supplement 1*). Similar to the observation made for mEPSCs, the decay kinetics of these evoked EPSCs were significantly different, with *Shisa7* KO currents showing a smaller decay tau (16.9 ± 1.4 *vs.* 12.0 ± 0.9, MWU-test, p=0.006; *Figure 5—figure supplement 1*). In

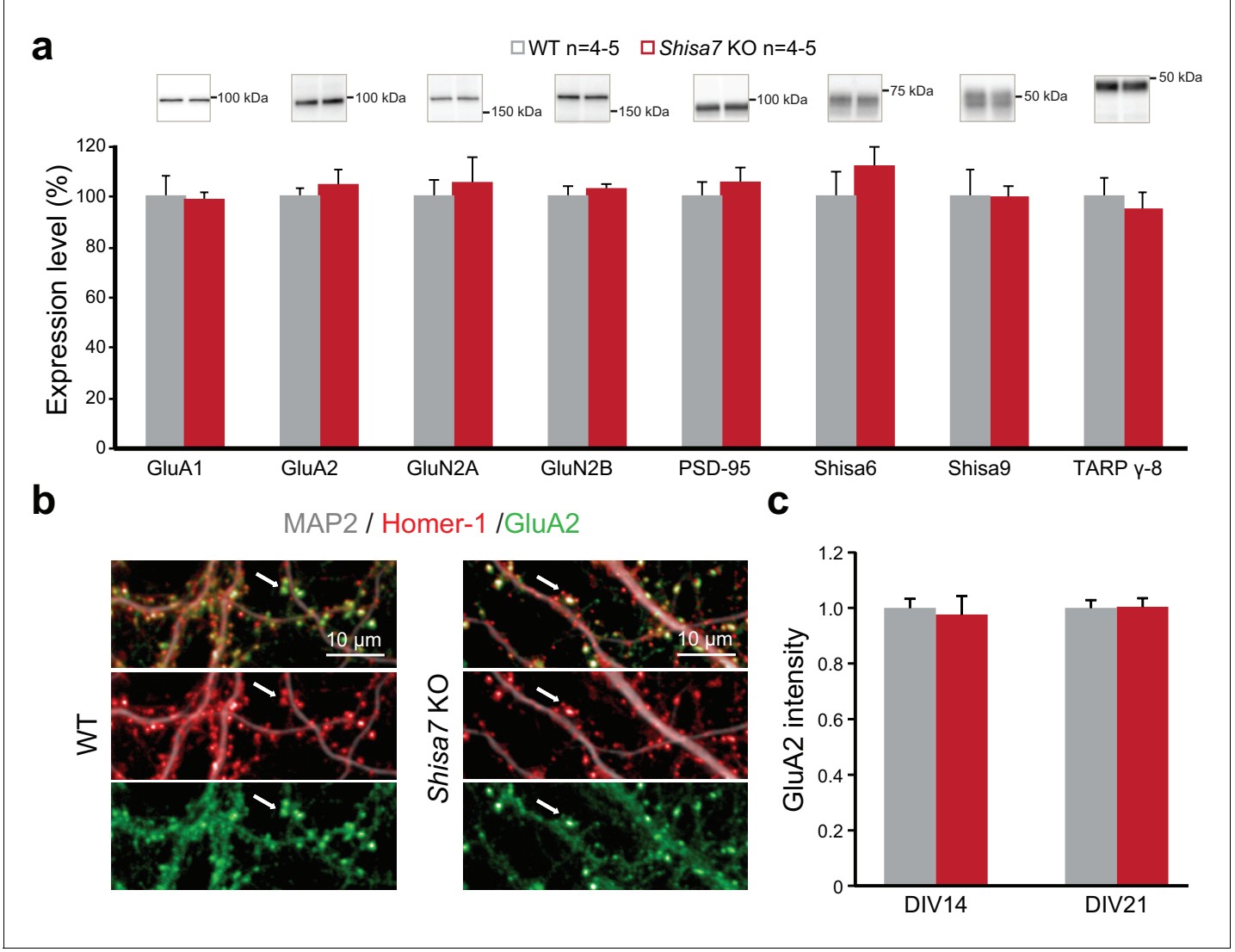

**Figure 4.** Deletion of Shisa7 maintains glutamatergic synapses under basal conditions. (a) Immunoblots of hippocampal synaptic membrane fractions from WT and *Shisa7* KO mice (n = 4–5 each) do not reveal differences in abundance of AMPAR (MWU tests, GluA1, p=0.421; GluA2, p=1.000), NMDAR (GluN2a, p=0.841; GluN2b, p=0.841), PSD-95 (p=0.841), TARP (γ−8, p=0.841), Shisa9 (p=0.421) or Shisa6 (p=0.310), when expressed as fold change over WT samples. The signal was normalized to the total protein content. (b,c) To assess the quality of the glutamatergic synapse, the intensity of the AMPAR GluA2 subunit was measured (example b; c; n = 10 wells, with five wells from two independent cultures). There was no genotype difference observed (DIV14, p=0.759; DIV21, p=0.919). Scale bars are indicated (b). For total protein loading used for normalization, see *Figure 4—figure supplement 2*.

DOI: https://doi.org/10.7554/eLife.24192.014

The following figure supplements are available for figure 4:

**Figure supplement 1.** Data as presented in *Figure 4*, but now with individual data points for WT (gray) and Shisa7 KO (red).
DOI: https://doi.org/10.7554/eLife.24192.015

**Figure supplement 2.** For immunoblots presented in *Figure 4*, we performed normalization of loading differences based on trichloroethanol-assisted total protein staining of the gel.
DOI: https://doi.org/10.7554/eLife.24192.016

addition, a faster rise time (2.80 ± 0.26 *vs.* 2.00 ± 0.13; MWU-test, p=0.023; *Figure 5—figure supplement 1*) was observed in the *Shisa7* KO. This effect was specific for the AMPAR, as the amplitude, rise and decay time was similar between *Shisa7* KO and WT animals for evoked NMDAR currents, while blocking AMPAR currents (*Figure 5—figure supplement 1*).

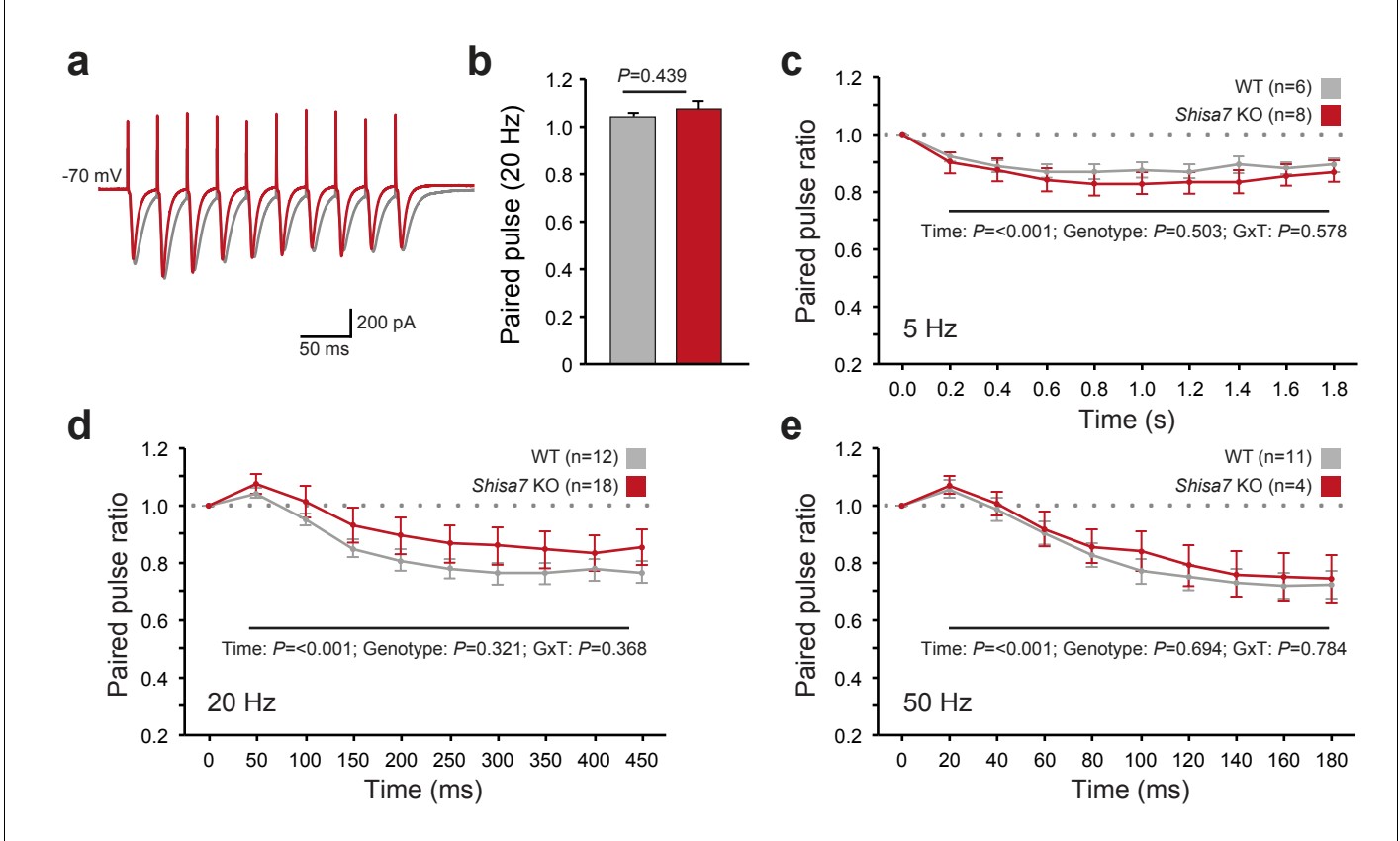

**Figure 5.** Shisa7 has no effect on short-term plasticity. (a) Superimposed example traces of whole-cell recordings voltage clamped at –70 mV from CA1 pyramidal neurons of *Shisa7* KO animals (red) and WT littermates (grey) in response to 50 Hz stimulation of synaptic inputs from Schaffer collateral fibers. (b) Paired pulse ratio is not affected in *Shisa7* KO animals, as exemplified by the 20 Hz data (WT: 1.04 ± 0.02; *Shisa7* KO: 1.07 ± 0.03; p=0.439). (c–e) Pulse ratios of electrically-evoked EPSCs from CA1 pyramidal neurons (at –70 mV) of *Shisa7* KO animals and WT littermates at 5 Hz (c), 20 Hz (d) and 50 Hz (e). At 5, 20 and 50 Hz, a clear desensitization was observed (factorial repeated measure ANOVA, time: F(2.97,35.65) = 18.21, p<0.001; F (1.55,43.39) = 35.55, p<0.001; F(1.69,21.93) = 35.529, p<0.001, respectively). At none of these frequencies there was a genotype effect or genotype x time interaction (all p>0.3). Cell numbers (n) used are indicated.

DOI: https://doi.org/10.7554/eLife.24192.017

The following figure supplement is available for figure 5:

**Figure supplement 1.** Similar AMPAR and NMDAR amplitudes in *Shisa7* KO mice.

DOI: https://doi.org/10.7554/eLife.24192.018

AMPAR desensitization properties can affect short-term synaptic plasticity and both Shisa9 and Shisa6 alter short-term synaptic plasticity in the hippocampus (*von Engelhardt et al., 2010*; *Klaassen et al., 2016*). Therefore, we tested whether Shisa7 affected the role of AMPARs in frequency-dependent short-term synaptic plasticity. Schaffer collaterals were stimulated at different frequencies to repeatedly activate glutamatergic inputs to CA1 pyramidal neurons (*Figure 5a*). We did not observe a difference in the paired-pulse ratios at any stimulation frequency (e.g., 20 Hz; MWU-test, p=0.439; *Figure 5b–e*). Similarly, with stimulation trains of 10 pulses at 5, 20 or 50 Hz (*Figure 5c–e*), we did not observe a change in synaptic depression. Thus, unlike Shisa9 and Shisa6, Shisa7 does not affect short-term synaptic plasticity.

### *Shisa7* gene deletion reduces long-term potentiation

Whereas TARP γ−8 is necessary for long-term potentiation of hippocampal glutamatergic synapses (*Rouach et al., 2005*), with specifically phosphorylation of S277/S281 being important for LTP as well as associative learning (*Park et al., 2016*), Shisa9 does not have such a role (*Khodosevich et al., 2014*). To investigate whether Shisa7 affects long-term synaptic plasticity, we

recorded from pyramidal neurons from CA1 hippocampal slices of *Shisa7* KO and WT mice and induced LTP of Shaffer collateral inputs at the stratum radiatum (*Figure 6*; see Materials and methods)(*Buzsáki et al., 1987*; *Buzsáki, 2005*). *Shisa7* KO mice showed a decrease in the maintenance phase of hippocampal LTP compared with WT littermates (*Figure 6a*; *Figure 6—figure supplement 1*). Synaptic potentiation was lower in *Shisa7* KO animals between 20 to 25 minutes (WT (n = 5): 1.81 ± 0.16; KO (n = 6): 1.10 ± 0.11; unpaired t-test, p=0.004), and 25 to 30 minutes (WT: 1.84 ± 0.16; KO: 1.17 ± 0.06; unpaired t-test, p=0.004; *Figure 6b*, *Figure 6—figure supplement 2*). In addition, the initiation of LTP was also significantly affected as apparent in the first 5 minutes after stimulation (0–5'; WT (n = 5): 2.46 ± 0.41; KO (n = 6): 1.52 ± 0.17; unpaired t-test p=0.049). In fact, during the first three 10 s measurements immediately after LTP induction, a genotype effect was detected (unpaired t-tests, p=0.057; p=0.034; p=0.077, respectively; *Figure 6c*). Compared with the first 5 minutes of their baseline recordings, both genotypes showed induction of LTP in the 5 minutes following theta burst stimulation (*Figure 6a*), however, this was more pronounced in WT animals (paired t-test, WT: p=0.004; KO: p=0.010). Similar results concerning maintenance and initiation of LTP were obtained when analyzing EPSP slope (*Figure 6—figure supplement 1*).

To study the role of Shisa7 in AMPAR recruitment, we induced LTP in primary neuronal hippocampal culture (DIV14–16) of WT and *Shisa7* KO mice (*Figure 6d*; *Figure 6—figure supplement 2*). At ~15 minutes after a brief application of glycine in WT neurons, the number of GluA1 spots was significantly increased at the surface compared with non-stimulated cells (t-test; p<0.001, *Figure 6e*; *Figure 6—figure supplement 2*). In line with the electrophysiological data, in *Shisa7* KO neurons the number of surface GluA1 spots was not increased, suggesting that recruitment of AMPARs at the surface was hampered (MWU test, p=0.894, *Figure 6e*; *Figure 6—figure supplement 2*).

## *Shisa7* KO mice show impaired contextual fear-conditioned memory

Because Shisa7 is important for initiation and essential to maintain LTP, we tested whether hippocampus-dependent learning is affected by analyzing contextual memory for an aversive stimulus (0.7 mA, 2 s shock). Two separate groups were tested, one in which expression of memory in terms of freezing behavior was assessed shortly after conditioning (2 hr), and one assessing long-term memory (24 hr). *Shisa7* KO mice showed a deficit in memory expression both at 2 hr (WT (n = 8): 45.3 ± 2.9%; KO (n = 7): 14.0 ± 2.86%; unpaired t-test, p<0.001), and at 24 hr after conditioning (WT (n = 9): 33.0 ± 5.2%; KO (n = 10): 9.2 ± 1.8%; MWU test, p<0.001) (*Figure 7a–d*, *Figure 7—figure supplement 1*).

To test whether *Shisa7* KO mice show different behavior either during the 3 minute baseline or during the 2 s shock acquisition that might explain a difference in US-CS acquisition, their mean locomotion velocity was analyzed (*Figure 7—figure supplement 2*). There was no genotype difference in base line locomotor activity (unpaired t-tests p=0.245) or shock perception (unpaired t-tests p=0.573). Together with locomotor data in a novel environment, i.e., distance moved in the open field test, our data indicate that *Shisa7* KO mice did not display altered locomotor activity (unpaired t-test p=0.106; *Figure 7—figure supplement 2*) or a difference in sensing the shock that might have confounded the acquisition and expression of fear memory.

*Shisa7* shows a widespread expression pattern throughout the brain, including the central nucleus of the amygdala (CeA; see *Figure 1—figure supplement 1*). Given the importance of the CeA in consolidation and expression of fear memory (*Wilensky et al., 2006*; *Ciocchi et al., 2010*), plasticity-related deficits by deletion of *Shisa7* might disturb US-CS acquisition and subsequent expression of fear, independently of hippocampal plasticity. Therefore, *Shisa7* KO and WT mice were tested in auditory fear conditioning, in which the tone that co-terminates with the shock is the foreground CS (*Phillips and LeDoux, 1994*). After conditioning, mice were tested in a novel context, in which they showed low levels of freezing (WT (n = 13): 10.7 ± 2.4%; KO (n = 11): 13.9 ± 2.9%; unpaired t-test, p=0.385)). Upon presentation of the tone, both KO and WT mice showed increased levels of freezing (WT: 31.7 ± 6.6%; KO: 41.3 ± 8.8%; *Figure 7e,f*) (factorial repeated measure ANOVA for switching on the tone, F(1,22)=27.80; p<0.001), with no genotype difference (F(1,22)=0.98; p=0.334), and lack of tone x genotype interaction (F(1,22)=0.47; p=0.500; *Figure 7—figure supplement 1*).

Finally, *Shisa7* KO mice were tested in several anxiety-related tasks, such as open field, elevated plus maze and dark-light box, in which no significant genotype effect was observed (*Figure 7—figure supplement 2*). Thus, we conclude that mice lacking the Shisa7 protein show reduced

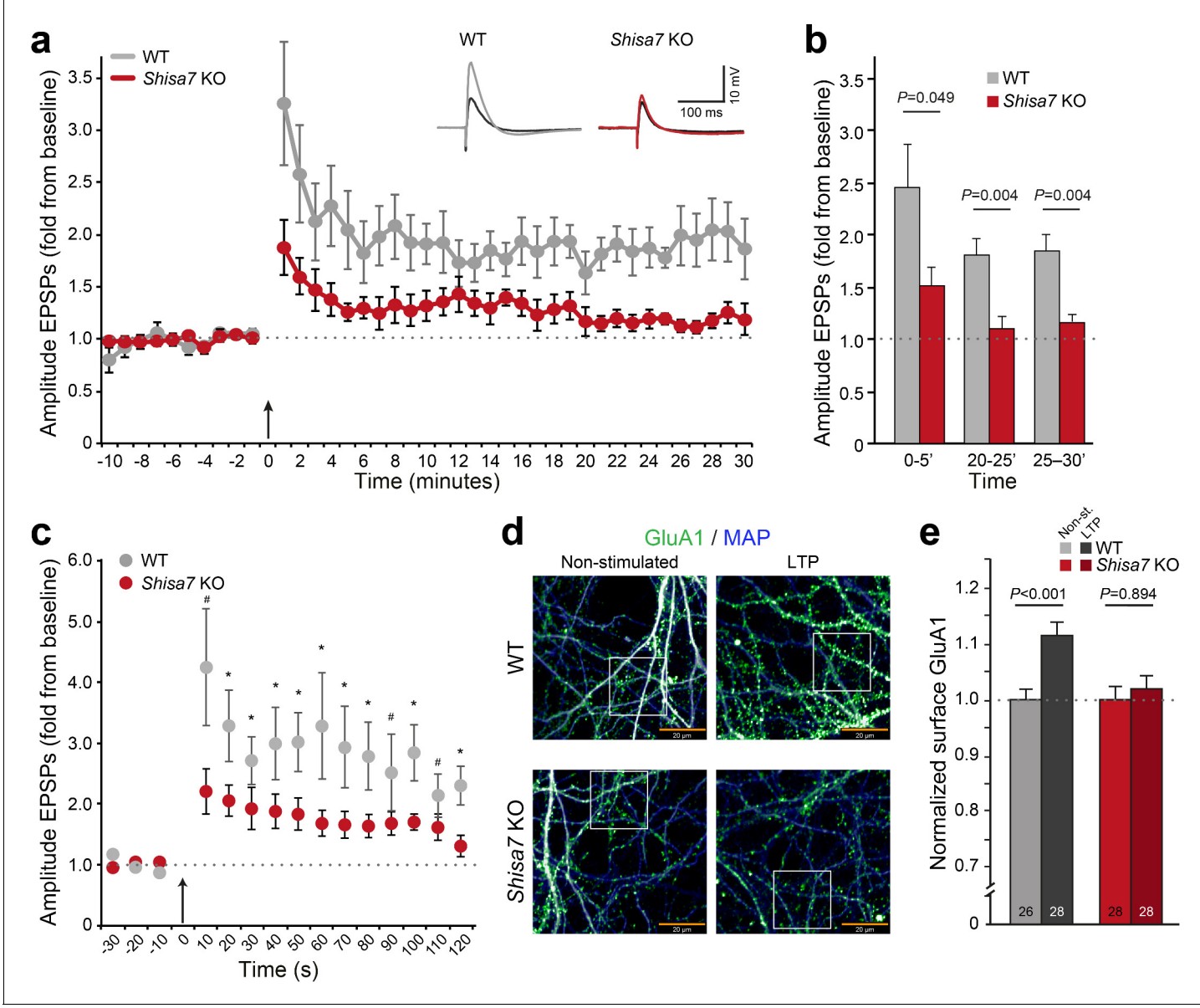

**Figure 6.** Deletion of Shisa7 slows down initiation and decreases maintenance of LTP. (a) Normalized EPSP amplitude over the time course of the LTP experiments shows a clear genotype effect (WT, n = 5 slices, n = 5 mice; *Shisa7* KO, n = 6 slices, n = 6 mice). The arrow indicates the theta burst stimulation. *Inset*: Example of LTP effect on WT (left) and the *Shisa7* KO (right) showing the EPSP shape during baseline (black) and after LTP induction (gray, red; last 10 minutes). (b) Normalized EPSP amplitude binned during 5 minute intervals early after LTP induction (0–5') and in the maintenance phase (20–25', 25–30') shows a significant genotype effect. Individual data are shown in *Figure 6—figure supplement 2*. (c) Zoom-in of the first 2 minute after theta burst application (arrow) for normalized EPSP amplitude showed an immediate genotype effect (unpaired t-tests, *p<0.05; #p<0.1). Dotted lines show base-line EPSP amplitude. (d) Example of GluA1 surface staining (green; MAP, blue) of hippocampal neurons (WT, *Shisa7* KO) after glycine-induced chemical LTP *vs.* non-stimulated (scale bar 20 μm). Insets are shown in *Figure 6—figure supplement 2*. (e) Quantification of number of surface GluA1 spots normalized to dendritic length from two independent experiments (n = 26–28 wells per group, based on average of 15–40 pictures per well) showed a significant effect of LTP in WT but not in *Shisa7* KO neurons. Individual data are shown in *Figure 6—figure supplement 2*.
DOI: https://doi.org/10.7554/eLife.24192.019

The following figure supplements are available for figure 6:

**Figure supplement 1.** Deletion of *Shisa7* slows down initiation and decreases maintenance of LTP.
DOI: https://doi.org/10.7554/eLife.24192.020

**Figure supplement 2.** Deletion of Shisa7 slows down initiation and decreases maintenance of LTP by affecting AMPAR recruitment.
DOI: https://doi.org/10.7554/eLife.24192.021

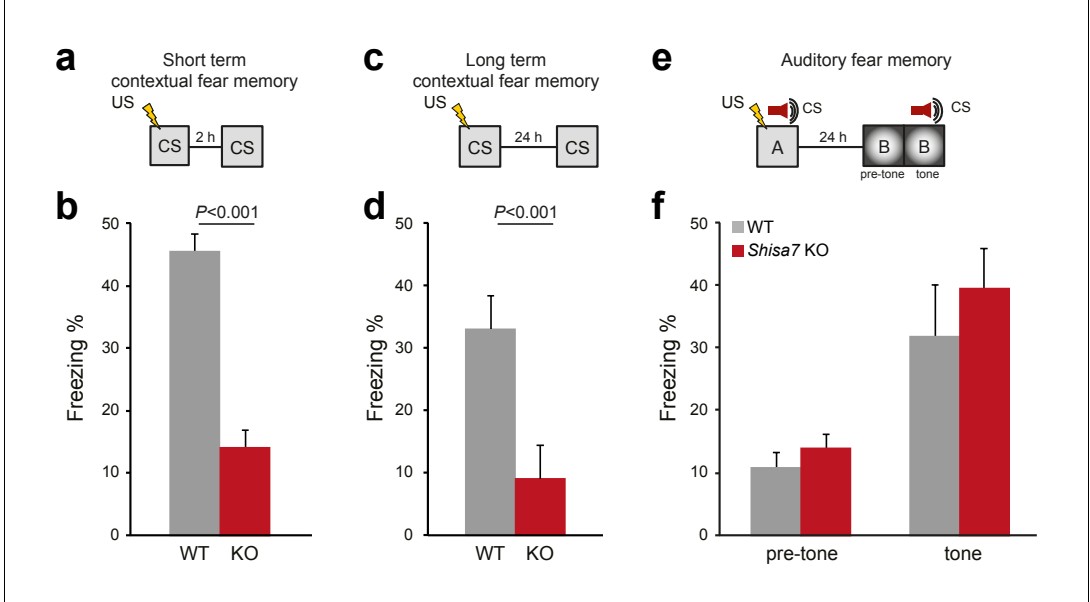

**Figure 7.** Deletion of *Shisa7* specifically affects contextual fear memory. (**a,c**) Experimental set-up of measuring contextual fear conditioning memory (**a, c**), in which mice received a foot shock (US) in a specific environment (CS), and freezing was assessed upon re-exposure to the CS 2 hr, or 24 hr later to measure short term and long term contextual fear memory, respectively. (**b,d**) For *Shisa7* KO mice, both a short term fear memory deficit ($n_{WT}$ = 8, $n_{KO}$ = 7; unpaired t-test, p<0.001; (**b**), as well as a long term fear memory deficit ($n_{WT}$ = 9 $n_{KO}$ = 10; Mann-Whitney U-test, p=0.001; (**d**) were observed. (**e**) Experimental set-up of measuring auditory fear conditioning memory, in which a 30 s tone co-terminated with a foot shock and memory was tested 24 hr after conditioning in a novel environment to measure generalization of fear (pre-tone), and the auditory fear memory in response to cue presentation (tone). (**f**) Auditory fear memory was not affected by genotype ($n_{WT}$ = 13, $n_{KO}$ = 12; ANOVA, $F_{(1,23)}$=0.733 p=0.401). Individual data is shown in *Figure 7—figure supplement 1*.

DOI: https://doi.org/10.7554/eLife.24192.022

The following figure supplements are available for figure 7:

**Figure supplement 1.** Deletion of *Shisa7* specifically affects contextual fear memory.

DOI: https://doi.org/10.7554/eLife.24192.023

**Figure supplement 2.** *Shisa7* KO mice display no abnormalities in shock sensation, locomotor activity or anxiety-related behavior.

DOI: https://doi.org/10.7554/eLife.24192.024

acquisition and expression of a contextual fear memory, but no difference in auditory fear learning or anxiety.

## Discussion

We identified Shisa7 as part of native hippocampal AMPAR complexes with unique characteristics compared with the Shisa9 and Shisa6 members of this protein family (*Figure 8*; *Supplementary file 1* [*Klaassen et al., 2016*]). Physical association of Shisa7 with the pore-forming GluA proteins modulates receptor properties in vitro by increasing AMPAR desensitization, lowering the steady state current and slowing down recovery of desensitization. Ex vivo, gene deletion of *Shisa7* affects rise and decay time of spontaneous and stimulus-evoked AMPAR currents in the hippocampus. Shisa7 displays a strong biochemical enrichment and localization at the post-synaptic density, and interacts with the scaffold protein PSD-95. Lastly, Shisa7 is likely involved in experience-dependent changes in synaptic AMPAR expression, as AMPAR recruitment during LTP as well as hippocampus-dependent expression of associative memory are strongly affected in *Shisa7* KO mice.

### Shisa7 is part of a postsynaptic AMPAR complex

Whereas Shisa9 is expressed most prominently in the hippocampus dentate gyrus (DG), Shisa6 (*Klaassen et al., 2016*) and Shisa7 are highly expressed throughout the hippocampus, in DG as well as CA1 to CA3 areas. Furthermore, Shisa7 has a wider expression pattern than Shisa6 and Shisa9. Shisa7 is found highly expressed in telencephalic structures (e.g., cortex, hippocampus, striatum),

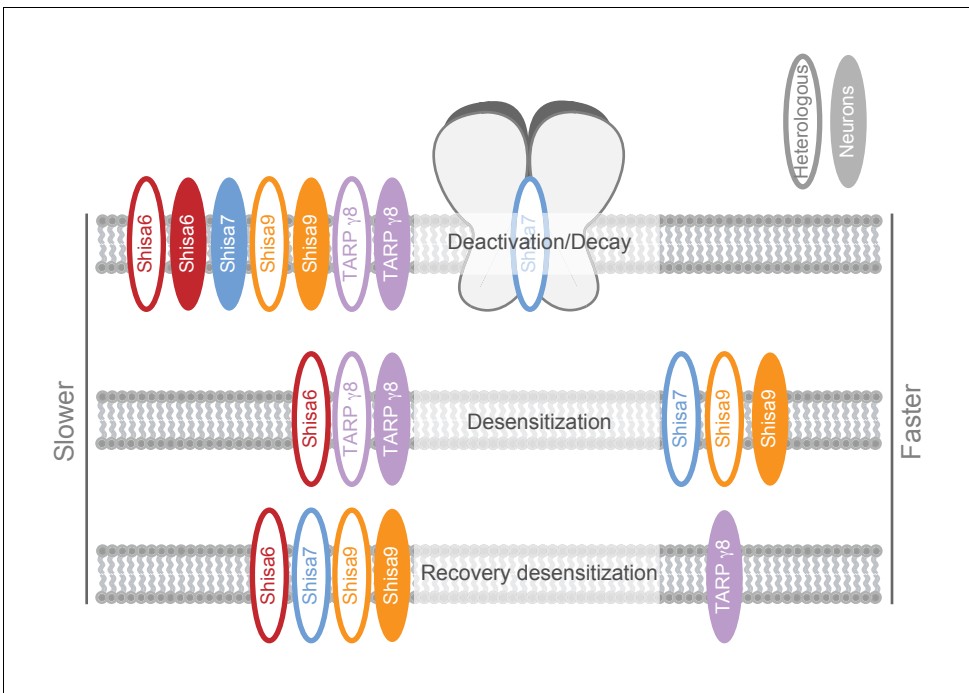

**Figure 8.** Schematic representation of effects on AMPA type glutamate receptors by Shisa family members and TARP γ−8. The effect is depicted for different AMPAR kinetic parameters measured in heterologous expression systems (HEK293 cells or oocytes; open ovals), or ex vivo in hippocampal slices (filled ovals). Additional modulatory properties and references are given in *Supplementary file 1*.
DOI: https://doi.org/10.7554/eLife.24192.025

and to a lesser extent in diencephalic (e.g., nuclei of the (sub- and hypo-) thalamus) and in some mesencephalic structures (e.g., superior and inferior colliculus), as can be observed at the transcript and in part at the protein level (*Figure 1*, *Figure 1—figure supplement 1*).

Like Shisa6 and Shisa9, Shisa7 is a postsynaptic protein that shows strong co-localization with Homer1 in hippocampal neurons and a high level of enrichment in the biochemically isolated PSD fraction (*Klaassen et al., 2016*). Shisa7 was found to interact with postsynaptic scaffolding protein of the disks large (Dlg) homolog family, PSD-95 (*Table 1*), via its C-terminal PDZ motif. In addition, immunoprecipitation experiments identified the established AMPAR auxiliary subunit Shisa6 as part of hippocampal Shisa7 complexes (*Table 1*). The synaptic context in which Shisa7 resides together with these other AMPAR interactors may enable the combined and possibly cooperative modulation of AMPAR properties.

Of the Shisa family members only Shisa6, and not Shisa9, was identified as co-interactor in the Shisa7-AMPAR complex. Thus, Shisa9 is not likely to decorate the same AMPAR population as Shisa7. Shisa6 was recently established as an AMPAR-auxiliary protein involved in AMPAR kinetics and AMPAR mobility (*Klaassen et al., 2016*). Moreover, Shisa6 was shown to interact with TARP γ−8. Although TARP γ−8 might not be a component of the Shisa7-AMPAR complex, as apparent from the immunoprecipitation in WT *vs. Shisa7* KO (*Table 1*), it may have gone undetected. Interestingly, the levels of other AMPAR auxiliary proteins did not change upon deletion of Shisa7, suggesting that receptors without Shisa7 are not perturbed in their remaining auxiliary protein composition.

## Shisa7 provides distinct modulation of AMPAR kinetics

Shisa7 is a Shisa family member with a distinct profile of AMPAR modulation (*Figure 8*; *Supplementary file 1*). Shisa7 influences AMPAR kinetics when co-expressed in HEK293 cells, resulting in an increase in the rate of desensitization, a lower sustained current in the prolonged presence of glutamate, and a reduced recovery from desensitization in contrast to what has been observed previously (*Farrow et al., 2015*). However, this opposition might reflect a difference in the Shisa7

and GluA1 protein levels, or a ratio thereof, upon expression in HEK293 cells. We found no effect on the rate of AMPAR deactivation. Shisa6 on the other hand, while bestowing a similar reduction in the AMPAR recovery from desentization, was found to reduce the rate of deactivation, reduce the rate of desensitization, and strongly increased the sustained steady state current (*Klaassen et al., 2016*). Interestingly, whereas all Shisa family members reduce the rate of recovery from desensitization, only TARP γ−8 increases the recovery time. Thus, albeit that Shisa7 is structurally more related to Shisa6 (*Pei and Grishin, 2012*; *Farrow et al., 2015*; *Klaassen et al., 2016*), some of its functional effects on AMPAR kinetics measured in a heterologous system resemble that of Shisa9, i.e. reducing sustained AMPAR current and increasing desensitization. This yields a new member of the Shisa family with unique properties.

Importantly, in contrast to the HEK293 cell system, when assessed within the neuronal context of hippocampal slices, Shisa7 deletion strongly shortens the decay time of mEPSCs and the decay and rise time of evoked AMPAR currents (*cf. Figures 3 and 5*), leaving NMDAR currents unaltered. This ex-vivo effect on evoked and/or miniature EPSC decay kinetics is a feature shared by most of the identified AMPAR auxiliary subunits, and has been reported for Shisa6 (*Klaassen et al., 2016*), Shisa9 (*von Engelhardt et al., 2010*), TARP γ−8 (*Khodosevich et al., 2014*), CNIH2/3 (*Boudkkazi et al., 2014*), but not for GSG1L (*Gu et al., 2016*). Since AMPAR deactivation was not affected upon co-expression of Shisa7 and GluA1-containing AMPARs in heterologous cells, we conclude that the observed alterations in AMPAR decay time in *Shisa7* KO neurons relies on the neuronal cellular context. These could be mediated by: (i) interactions of Shisa7 with GluA2- and GluA3-subunit containing AMPARs, (ii) neuronal phosphorylation states of the receptor or Shisa7, (iii) (secondary) interactions of Shisa7 with other auxiliary proteins in the complex resulting in a combined and possibly cooperative modulation of AMPAR kinetics, or (iv) *Shisa7* deletion rendering the AMPAR in a state that allows other auxiliary subunits to more prominently modulate biophysical properties of the receptor. In the latter case, it must be noted that no compensatory changes were observed in the synaptic membrane expression levels of AMPAR auxiliary subunits TARP γ−8, Shisa6 or Shisa9 in the *Shisa7* KO condition. Finally, testing of the aforementioned ideas would require a technically challenging reconstitution of the protein complexes in vitro, based on the exact complex composition identified in vivo, to search for a mimic of the physiological effects that were observed in brain slices.

Interestingly, the effect of AMPAR auxiliary subunits on short-term synaptic plasticity appears to be diverse, even within the same protein family. Whereas Shisa6 and TARP γ−8 (*Rouach et al., 2005*) facilitate short-term plasticity, Shisa9 decreases short-term plasticity (*von Engelhardt et al., 2010*; *Khodosevich et al., 2014*), and Shisa7 is without effect. Apparently, there is no direct correlation or causality between the effect of Shisa and TARP proteins on in vitro AMPAR recovery from desensitization and ex vivo synaptic depression.

## Shisa7 is required for synaptic plasticity-dependent changes in AMPAR expression

Most of the yet identified AMPAR-auxiliary proteins, such as Shisa9, members of the Cornichon and TARP family control AMPAR surface expression (*Rouach et al., 2005*; *von Engelhardt et al., 2010*; *Coombs et al., 2012*; *Gill et al., 2012*). Here we show that the absence of Shisa7 has no effects on the amplitudes of mEPSCs and electrically-evoked amplitudes in hippocampal CA1 neurons, in line with results obtained for Shisa6 (*Klaassen et al., 2016*). These data imply that both Shisa6 and Shisa7 are not involved in synaptic recruitment of AMPARs under basal conditions. We further confirmed this by showing that the expression of the AMPAR in biochemically isolated synaptic membrane fractions, and in synapses of primary hippocampal cultured neurons is not altered due to Shisa7 deletion.

Although basal expression of the AMPAR is not dependent on Shisa7, plasticity experiments revealed that induction and maintenance of LTP in the Schaffer collaterals fails dramatically in *Shisa7* KO mice. Firstly, deletion of *Shisa7* strongly affects the initiation phase of LTP, as is most profoundly visible when analyzing EPSP amplitudes (*Figure 6*, *Figure 6—figure supplement 1*). This initial phase of LTP is characterized by an increase in intracellular calcium (*Borgdorff and Choquet, 2002*), followed by a reduction in AMPAR mobility (*Heine et al., 2008*; *Petrini et al., 2009*) through diffusional trapping, a process that is dependent on TARP phosphorylation (*Opazo et al., 2010*). Despite the fact that the Shisa family members do not contain the typical class I PDZ binding motif found in

Dlg-interacting proteins but rather the type II (*Songyang et al., 1997*), our data reveal that Shisa7 is directly associated with the prominent diffusion slot organizer and postsynaptic scaffold protein PSD-95 (DLG4) through its C-terminal PDZ-motif. A similar PDZ-protein association was observed for Shisa9 (*Karataeva et al., 2014*) and Shisa6 (*Klaassen et al., 2016*). Specifically for Shisa6, we previously showed that this interaction underlies its ability to reduce AMPAR mobility. However, we here did not perform these experiments, mainly due to the observed loss of spines after Shisa7 overexpression. Taken together, it is suggested that Shisa7 plays a role in immobilization of the AMPAR in the PSD upon stimulation, likely due to binding of synaptic scaffolds or via indirect modulation of TARP phosphorylation.

The subsequent phases of LTP are characterized by the exocytosis and recruitment of additional AMPARs to postsynaptic sites (*Jurado et al., 2013*), thereby providing a lasting increase in synaptic strength and spine size (*Liao et al., 1995*; *Kopec et al., 2007*). *Shisa7* KO animals show a profound deficit in this maintenance phase of LTP, yet they do not display impaired synaptic AMPAR targeting under basal conditions. This phenotype was corroborated in cultured neurons, in which the chemical LTP-induced increase in surface GluA1 was not observed in the *Shisa7* KO.

The *Shisa7* KO LTP phenotype in the CA1 region resembles that of TARP $\gamma-8$ KO mice (*Supplementary file 1* [*Rouach et al., 2005*]). On the other hand, Shisa9 has no effect on LTP, as measured in the dentate gyrus (*von Engelhardt et al., 2010*; *Khodosevich et al., 2014*), but it does modulate AMPAR surface expression similar to TARP $\gamma-8$. Together, this highlights the unique role of Shisa7 on AMPAR functionality compared with other Shisa proteins. We propose that in contrast with other auxiliary subunits, such as Shisa6, Shisa7 has only mild effects on AMPAR gating and therefore does not influence short-term plasticity. Indirect interaction with the protein machinery that is required for AMPAR exocytosis (*Jurado, 2014*) could aid to maintain synapses in a potentiated state, as is necessary for LTP. Furthermore, after LTP induction, Shisa7 could act as scaffolding protein in the PSD through direct interaction with Dlg family members to promote diffusion trapping based on its C-terminal type II PDZ-ligand motif (EVTV). A rescue experiment introducing Shisa7 with and without this PDZ-ligand motif in a *Shisa7* KO background could support this hypothesis. However, considering that the level of Shisa7 expression through viral delivery cannot be controlled and viral Shisa7 expression leads to spine loss, this experiment would lead to inconclusive results.

## Shisa7 is specifically involved in plasticity underlying contextual memories

Although AMPAR auxiliary subunits have been studied in detail for cellular hippocampal plasticity processes, studies on the biological significance in terms of general behavioral effects and cognition are sparse and so far have been studied for TARP $\gamma-8$ (*Gleason et al., 2015*; *Park et al., 2016*) and GSG-1L (*Gu et al., 2016*). The *GSG-1L* KO rat has no spatial memory deficits in the Morris Water Maze and a spatial object recognition test, although non-spatial novel object recognition memory is impaired. AMPAR-directed compounds that partially disrupt the interaction between TARP $\gamma-8$ and the AMPAR prevent the effect of TARP $\gamma-8$ on desensitization and deactivation, and have a dualistic effect when systemically administered during spatial learning (*Maher et al., 2016*); at high dose it impairs acquisition of spatial memory in the Morris water maze, whereas at low doses it has a facilitating effect. In both cases, there is no effect on the subsequent probe trial testing spatial recollection memory. Recently, the role of CaMKII-dependent phosphorylation of TARP $\gamma-8$ has been studied, and mutation of S277/S281 leads to decreased contextual and auditory fear memory (*Park et al., 2016*).

In *Shisa7* KO mice, we observed a specific impairment in the expression of contextual fear memory only. The disrupted expression of contextual memory both at 2 hr and 24 hr after acquisition points to a deficit in consolidation, and possibly to the acquisition of the US-CS association. Integrity of the dorsal hippocampus is necessary for the formation and retrieval of contextual memories, but not for auditory fear conditioning in both rats and mice (*Kim and Fanselow, 1992*; *Phillips and LeDoux, 1994*; *Maren et al., 1997*; *Stiedl et al., 2000*). In addition, the basolateral amygdala contributes both to the contextual component of fear memory retrieval, as well as to the cued component of tone-shock pairings (*Gale et al., 2004*). It is of interest that only contextual memory was impaired, leaving auditory fear memory intact, whereas Shisa7 is expressed in the hippocampus and amygdala at both the transcript and protein level. Hence, the impact of Shisa7 deletion on plasticity

is probably brain region-dependent, which might result from the distinct composition of the associated proteins that reside in different Shisa7-AMPAR complexes.

## Conclusion

We showed that *Shisa7* is functional in AMPAR-containing synapses in the hippocampus and is involved in the fine-tuning of the channel properties of AMPARs, affects hippocampal synaptic plasticity, and takes part in hippocampus-dependent fear memory. The current study demonstrates the concurrent involvement of multiple AMPAR-associated proteins in the physiological and behavioral phenotype, and argues for the molecular dissection of highly composite AMPAR complexes, and their function in specific brain areas.

# Materials and methods

## Mice

Mice were bred in the facility of the VU University Amsterdam. They were group-housed in standard type-2 Macrolon cages enriched with nesting material on a 12/12 hr rhythm (lights on at 7:00 AM). The housing area had a constant temperature of 23 ± 1 °C and a relative humidity of 50 ± 10%. Food and water was provided ad libitum. All the behavioral experiments were performed between 9:00 AM and 5:00 PM. Protein and RNA samples were prepared from 8 to 14 week old male and female C57/BL6J mice, derived from Charles River (France). Immunoprecipitations were performed on hippocampi of 8–14 week old male and female WT and *Shisa7* KO mice. All electrophysiological recordings on CA1 neurons, as well as behavioral analyses, were performed on 8–12 week old male littermates. Male *Shisa7* KO and WT littermates were single-housed when adult (>8 weeks),~2 weeks prior to behavioral analysis; behavior was carried out with mice 10–12 weeks. All experiments were performed in accordance to Dutch law and licensing agreements using protocols approved by the Animal Ethics Committee of the VU University Amsterdam.

### Generation of Shisa7 KO mice

For generation of *Shisa7* KO mice, see *Figure 1—figure supplement 2*.

## (Real-Time) polymerase chain reaction

### Primers

Primers for PCR and real-time PCR were generated using Primer3.0. Primer sets are listed in *Figure 1—source data 1*.

### RNA isolation and cDNA synthesis

RNA from different tissues (pool of 3 adults) and several time points during development (n = 3 independent samples with two animals pooled; based on previous experience; (*Klaassen et al., 2016*)) was extracted as previously described (*Spijker et al., 2004*). Samples were DNase-I treated according to the manufacturer's instructions (20 U per µg RNA; Boehringer) to remove traces of genomic DNA, which was verified by using intron-specific PCR primers (data not shown). RNA concentration was determined using the NanoDrop ND-1000 spectrophotometer (NanoDrop Technologies; Wilmington, DE), and the integrity of RNA was checked by gel electrophoresis (1%-TBE-agarose gel). Random-primed (25 pmol; Eurofins MWG Operon; Ebersberg, Germany) cDNA synthesis was performed on individual RNA samples (~1 µg total RNA).

### PCR for exon 4

PCR reactions on two WT samples were generated with Ex2-Ex5/6 primers with 0.5 U Phusion (New England Biolabs; Hitchin, UK) in a 50 µL reaction using the HF buffer according to the manufacture's protocol.

### Real-time qPCR

Real-time qPCR reactions (20 µL; ABI PRISM 7700) were performed using a 96-well format with transcript-specific primers (300 nM) on cDNA corresponding to ~20 ng RNA, as described previously

(*Spijker et al., 2004*), using SYBR Green reagents (Applied Biosystems; Waltham, MA) (see supplementary information). Only primer sets (Eurofins MWG Operon) with proper amplification efficiency (*Jacobs et al., 2002*; *Spijker et al., 2004*) were used (*Figure 1—source data 1*). Cycle threshold (Ct) values were used to calculate the relative level of gene expression, where Ct value is the fractional cycle number at which the fluorescent signal of a reaction passes the threshold (reaching intensity above background). Expression level of three housekeeping genes (GAPDH, β-actin, HPRT) was measured as reference for input. Expression is denoted using normalized Ct values on a $\log_2$-scale. Let normalized Ct-values be denoted by $Ct\text{norm}_x$ (where $x$ represents expression the gene of interest, $y$ represents the geometric mean of Ct-values of the housekeeping genes, and $i$ represents a given sample), $Ct\text{norm}_{xi}$ then is given by $Ct\text{norm}_{xi} = Ct_{xi} - Ct_{yi}$. As a bigger Ct-value correlates with a lower expression level, for practical purposes, $Ct\text{norm}_{xi}$ values were converted into $con Ct\text{norm}_{xi}$ values, calculated as $con Ct\text{norm}_{xi} = -Ct\text{norm}_{xi} + 15$. Due to this conversion, the final positive value of Ct is positively correlated with relative gene expression level, which makes the visualization simpler. Relative gene expression levels were expressed as $con Ct\text{norm}$-values ±SEM.

## DNA expression constructs

The full-length coding DNA for exon4-containing mouse Shisa7 (reference NP_766325.3) was obtained by PCR amplification, using a mouse brain Matchmaker cDNA library (Clontech; Saint-Germain-en-Laye, France) as template, and Gateway-cloned into pTRCGw-IRES2-EGFP (*Klaassen et al., 2016*), yielding the Shisa7-pTRCGw-IRES2-EGFP construct. This plasmid was modified to FLAG-Shisa7-pTRCGw-IRES2-EGFP by PCR-mediated insertion of a tandem FLAG-tag (sense 5′-GGT GAT TAT AAA GAT CAT GAT ATC GAT TAC AAG GAT GAC GAT GAC AAG CAC-3′, corresponding peptide: GDYKDHDIDYKDDDDKH) between codon 28 (ACA, Threonine) and codon 29 (AGC, Serine) of the Shisa7 cDNA.

For HEK293 cell expression, constructs used are described previously (*Klaassen et al., 2016*).

## Antibodies

Anti-Shisa7 antibody was raised in rabbit against sequence GTLARRPPFQRQGT (position 519–532 in Shisa7) (Genscript; Piscataway, NJ). The antibody was affinity-purified against the antigenic peptide, suspended at 1 mg per mL in PBS containing 0.02% NaN3, and stored at –20 °C.

Antibodies used for immunoprecipitation were anti-Shisa7 (see above) and anti-FLAG M2 (Sigma-Aldrich, F1804; Zwijndrecht, Nederland).

Antibodies used for immunoblotting were anti-Shisa7 (see above, 1:1,000), anti-GluA1 (Abcam, ab109450, 1:20,000; Cambridge, UK), anti-GluA2 (Neuromab clone L21/32, 1:1,000; Davis, CA), anti-GluA3 (Abcam, ab40845, 1:500; Cambridge, UK), anti-GluK2 (Santa Cruz, C-18, 1:1,000; Heidelberg, Germany), anti-GluN2A (Abcam, ab14596, 1:2,000; Cambridge, UK), anti-Glun2B (Neuromab 1:1,000; Davis, CA), anti-PSD-95 (Neuromab clone K28/43, 1:20,000; Davis, CA), anti-Synaptophysin (Genscript A01307, 1:2,000; Piscataway, NJ), anti-Shisa6 (custom-made (*Klaassen et al., 2016*) 1:1,000), anti-Shisa9 (Santa Cruz, 1:500; Heidelberg, Germany), anti-TARP γ−2,–4, −8 (Neuromab clone N245/36, 1:500; Davis, CA). Horseradish peroxidase-conjugated secondary antibodies were obtained from Dako (1:10,000; Santa Clara, CA).

Antibodies used for immunocytochemistry were anti-MAP2 (MerckMillipore AB5543, 1:2,000), anti-GluA2 (Neuromab clone L21/32, 1:400; Davis, CA), anti-FLAG M2 (Sigma-Aldrich, F1804, diluted 1:1,000; Zwijndrecht, Nederland), Homer1 (Synaptic Systems 160 004, 1:1,000; Goettingen, Germany), GluA1-N (Merck Millipore, ABN241, 1:200). Alexa-conjugated isotype-specific secondary antibodies were obtained from ThermoFisher (1:400; Waltham, MA).

## Immunoblot analysis

Protein samples were dissolved in SDS sample-buffer (Laemmli), heated to 96 °C for 5 minutes, and loaded onto a 4–15% Criterion TGX Stain-Free gel (Bio-Rad; Temse, Belgium). The gel-separated proteins were imaged with the Gel-Doc EZ system (Bio-Rad; Temse, Belgium), transferred onto PVDF membrane (Bio-Rad; Temse, Belgium) and probed with various antibodies (see 'Antibodies' section). Scans were acquired with the Odyssey Fc system (Li-Cor; Lincoln, Nebraska), and adjusted using Image Studio Lite 5.2.5 software (Li-Cor; Lincoln, Nebraska). Immunoblot band intensities were normalized to the total amount of protein loaded as quantified using Image Lab 3.0 software (Bio-

Rad; Temse, Belgium). Sample size for quantitative immunoblotting in KO and WT samples was based on (*Klaassen et al., 2016*).

## Subcellular fractionation

Subcellular fractions were prepared as described previously (*Karataeva et al., 2014*) with some modifications (*Klaassen et al., 2016*).

## Immunoprecipitation of Shisa7 protein-complexes from mouse hippocampus

Hippocampal tissue from WT and *Shisa7* KO mice was prepared for immunoprecipitation using anti-Shisa7 antibody on the DDM-extracted crude synaptic membrane fraction, as previously described (*Klaassen et al., 2016*). Proteins were separated by SDS-PAGE and after in-gel digestion the peptides were subjected to mass spectrometer analysis (TripleTOF 5600+ system (Absciex) operated in Information Dependent Acquisition mode), as previously described (*Klaassen et al., 2016*).

TripleTOF 5600+ raw files were imported into MaxQuant (version 1.6.0.1) (*Cox and Mann, 2008*) and searched against the mouse UniProtKB/Swiss-Prot canonical sequence database (09–2017 release). Methionine oxidation and protein N-terminal acetylation were selected as variable modifications, and proprionamide was set as fixed cysteine modification. For both peptide and protein identification the false discovery rate was set to 0.01. MaxLFQ normalisation was enabled with a LFQ minimal ratio count of 1. Remaining parameters were left at default. The Maxquant results at proteingroup level are provided in *Table 1—source data 1*. Next, the statistical significance of the MaxQuant results was assessed by importing the proteinGroup.txt file into Perseus (version 1.6.0.2) (*Tyanova et al., 2016*) and conducting the following workflow: (1) Removal of 'Reverse', 'Potential contaminant', and 'Only identified by site' proteingroups; (2) Log(2) transformation of all LFQ intensity values; (3) Removal of proteingroups with less than three valid 'Log(2) LFQ intensity' values in either the WT or KO groups; (4) Imputation of missing values (8.6% of the population) from a normal distribution (width 0.3, down shift 1.8, whole matrix); 5) Performing a Student's t-test with permutation-based FDR analysis (S0 = 1, FDR = 0.01, 2500 permutations). The Perseus results are provided in *Table 1—source data 2*.

## Co-precipitation from HEK293 cells

For protein extraction, HEK293 cells were washed with PBS, resuspended in lysis-buffer (1% Triton X-100, 150 mM NaCl, 25 mM HEPES (pH 7.4), and EDTA-free Complete protease inhibitor), and incubated at 4 °C for 1 hr while mixing gently. The supernatant was cleared of non-soluble debris by two consecutive centrifugation steps at 20,000x *g* for 20 minutes. Anti-flag antibody was added to the supernatant, incubated O/N, and immobilized to Protein A/G agarose beads (Santa Cruz; Heidelberg, Germany). The agarose beads were washed four times with lysis buffer, and bound proteins were eluted by incubation with Laemmli sample buffer.

## Yeast two-hybrid

A direct two-hybrid assay was performed in PJ69-2 yeast cells between PSD-95 (amino acids 39–262 of NP_031890.1, encoding PDZ domains 1 and 2) and either the wild-type cytoplasmic domain of Shisa7A (Shisa7-cd WT, amino acids 210–558) or the truncated mutant thereof (Shisa7-cd ΔEVTV), as described previously (*Karataeva et al., 2014*). Cell-growth was recorded after 4 days of stringent nutritional selection (-Leu, -Trp, -His, -Ade), as described previously (*Karataeva et al., 2014*).

## Primary neuron culture

Hippocampi were dissected from E18 WT an *Shisa7* KO mice, collected in Hank's balanced salts solution (HBSS, Sigma-Aldrich, Zwijndrecht, Nederland) buffered with 7 mM HEPES (pH 7.4) and incubated for 30 minutes in HBBS containing 0.25% trypsin (Life technology; Waltham, MA) at 37 °C. After washing, neurons were triturated with fire-polished Pasteur pipettes, counted, and plated in Neurobasal medium supplemented with 2% B-27, 1.8% HEPES, 1% glutamax, 1% penicillin/streptomycin (all from Life technology; Waltham, MA), and 0.2% 14.3 mM β-mercapto-ethanol. Cultures were plated on 96-well cell culture microplates (Cellstar, Greiner Bio-One, Frickenhausen, Germany) coated in poly-d-lysine (Sigma-Aldrich; Zwijndrecht, Nederland) and treated with 5% heat-

inactivated horse serum (Invitrogen) at a seeding density of 1250 cells/mm$^2$ and kept at 37 °C/5% CO2. Neurons were fixed using 4% PFA, 1% sucrose in PBS (pH 7.4) for 20 minutes at DIV14 and DIV21.

## Immunocytochemistry

For quantification of the AMPAR expression in synapses, GluA2, Homer1, and MAP2 immunolabeling was performed. Wells with primary neurons (DIV14, 21) were washed with PBS and non-specific binding was blocked by incubation with blocking solution containing 0.2% Triton-X100, 3% BSA in PBS for 1 hr at RT. Primary antibody incubation was performed at 4 °C ON in blocking solution followed by additional washing with PBS. Secondary antibodies were incubated in blocking solution for 1 hr at RT, followed by a 10 minute Hoechst incubation (1:10,000, Thermo Fisher Scientific; Waltham, MA). For LTP-induced surface GluA1 labeling, live hippocampal neurons were incubated with anti-GluA1-N in a humidified incubator (37 °C / 5% CO$_2$) for 6–7 minutes. Neurons were then fixed with 4% paraformaldehyde/2% sucrose (20 minutes), followed by permeabilization with 0.5% Triton-X100 (10 minutes). Neurons were blocked with 2% BSA/0.1% Triton-X100 for 1 hr before O/N incubation in primary MAP2 antibody diluted in blocking solution. The following day, neurons were washed and incubated with anti-rabbit Alexa 568- and anti-chicken Alexa 647-conjugated secondary antibodies for 90 minutes at RT.

## Shisa7-FLAG expression and staining

For Shisa7-FLAG expression, primary neurons were plated on coverslips in 24-well culture plates (Cellstar, Greiner Bio-One, Frickenhausen, Germany) as described above. Shisa7-FLAG expression was induced by adding lentivirus to the cell medium at DIV7 (*pLenti-CMV-Shisa7FLAG-IRES-GFP*). Neurons were fixed at DIV18 with 4% PFA, 1% sucrose in PBS pH 7.4 for 20 minutes. Coverslips with GFP-expressing neurons were stained with anti-FLAG-M2 and Homer1 antibody. Fixed coverslips were washed in PBS and permeabilized with 0.2% Triton-X100 in PBS for 5 minutes. Non-specific binding was blocked by a 45 minute incubation with blocking solution containing PBS with 3% BSA and 0.1% Triton-X100. Primary antibody incubation was performed overnight at 4 °C in blocking solution, followed by thorough washing and secondary antibody incubation in blocking solution for 1 minute at RT. Coverslips were mounted on glass slices using Polyvinyl alcohol antifading mounting medium with DABCO (Sigma-Aldrich, Zwijndrecht, Nederland) Images were generated by confocal microscopy (LSM 510, Zeiss, Jena, Germany).

## Chemical LTP induction

At DIV6–7, half of the neuronal medium was replaced with equilibrated BrainPhys medium (*Bardy et al., 2015*). This was repeated at least two more times before chemical LTP treatment at DIV14–16. Neurons were stimulated by exchanging the BrainPhys medium with chemical LTP solution (in mM): 120 NaCl, 5 KCl, 20 HEPES, 15 Glucose, 2 CaCl$_2$, 0.2 Glycine, 0.03 Bicuculline methiodide, pH 7.4. Control neurons were treated with normal tyrode (mM): 120 NaCl, 5 KCl, 20 HEPES, 15 Glucose, 2 CaCl$_2$, 2 MgCl$_2$. Chemical LTP treatment took place in a humidified incubator (37°C/5%CO$_2$) for 3–4 minutes before gentle washing with PBS. This was quickly followed by live-labelling of GluA1-N for surface expression of GluA1, and subsequent fixation with 4% PFA 15 minutes after chemical LTP treatment.

## Image analysis

Expression levels of synaptic AMPARs in cultured neurons were analyzed by measuring GluA2 intensity levels in postsynaptic Homer1-positive spots. Hippocampal primary culture was performed in with 5 *Shisa7* KO and WT wells per plate, and intensity values were normalized to the average WT data per plate. Wells were scanned using confocal imaging at 60X magnification with an Opera High Content Screening System (PerkinElmer; Waltham, MA). In total 40 images per well were analyzed, with ~50 neurons per well. All images included in the analysis contained at least one nucleus. Postsynaptic spots (Homer1 positive) were detected in MAP2-positive neurite regions, and GluA2 intensity was measured in the postsynaptic spot region as a measure for synaptic GluA2 expression. For analysis of the images and quantification of synaptic GluA2 expression the Columbus image storage and data analysis system (PerkinElmer; Waltham, MA) was used. The experiment was independently

replicated (*Schreiber et al., 2015*) with a similar set-up from a different culture using independent sets of animals, and the resulting 2 × 5 wells were taken as the final n-number.

Similarly, for measuring surface GluA1 after chemical stimulation, confocal images were acquired with a 40x objective at a fixed focal plane (maximally 40 images per well) with the Opera High Content Screening system (PerkinElmer). Subsequent image analysis, to quantify neurite length and number of detected GluA1 spots, was performed with the Columbus software (version 2.5.2.124862, PerkinElmer). Density of surface GluA1 was calculated by taking the number of GluA1 spots over the sum of neurite length per well.

## HEK293 cell culture, transfection and electrophysiological recording

HEK293 cells (ATCC; Manassas, VA) were cultured in DMEM media (Gibco, Life Technologies; Waltham, MA) supplemented with 10% FBS (Invitrogen; Waltham, MA) and 1% Penicillin-Streptomycin (Gibco, Life Technologies; Waltham, MA). To ensure consistency in culture, HEK293 cells were passaged no more than 16 times. HEK293 cells were transfected with plasmids encoding glutamate receptor subunits (GluA1-3 and GluK2) alone or in combination with (flag) Shisa7 (see 'DNA expression constructs') using Polyethylenimine (PEI) (25 kDa linear, Polysciences; Hirschberg an der Bergstrasse, Germany) and incubated for 42–66 hr. Cells were 60–70% confluent at the time of transfection. Two hours prior to recording, HEK293 cells were transferred to coverslips coated with 100 µg/mL Poly-L-Lysine (Sigma Aldrich, Zwijndrecht, Nederland). All electrophysiological recordings were performed as described previously (*Klaassen et al., 2016*). Sample size was based on previous results (*Klaassen et al., 2016*).

## Electrophysiological recordings in acute hippocampal brain slices

Mice were decapitated and the brain was removed from the skull in ice-cold slice solution containing (in mM): 126 NaCl, 3 KCl, 10 D-glucose, 26 NaHCO$_3$, 1.2 NaH$_2$PO$_4$, 0.5 CaCl$_2$ and 7 MgSO$_4$ carboxygenated with 95% O$_2$ and 5% CO$_2$, pH 7.4. Acute horizontal hippocampal slices were cut with a thickness of 300 µm using a vibratome (Microm HM 650 V) in ice-cold slice solution and transferred to standard carboxygenated aCSF (2 mM CaCl$_2$ and 1 mM MgSO$_4$) for a recovery period of at least 1 hr prior to recordings. Unless indicated otherwise, salts were purchased at Sigma Aldrich (Zwijndrecht, Nederland) and drugs were purchased at Abcam (Cambridge, UK). Sample size was based on previous results (*Klaassen et al., 2016*).

Glass electrodes of 6–9 MΩ resistance were used for all whole-cell recordings from acute brain slices and pulled using borosilicate glass (OD 1.5 mm, ID 0.86 mm; Harvard Apparatus, Holliston, MA), and were filled with intracellular solution containing (in mM): 117 K-gluconate, 8 KCl, 10 HEPES, 4 Mg-ATP, 0.4 Na-GTP, 10 K-phosphocreatine, 0.2 EGTA, (pH 7.3, adjusted with KOH; osmolarity 290 mOsm/L). Slice recordings were performed using standard aCSF (see above) at 30–31 °C at a flow rate of 3–4 mL/minute.

### Miniature EPSCs

Whole-cell patch clamp recordings of CA1 pyramidal cells (voltage-clamped at −70 mV) were performed with an Axopatch 200B amplifier (Molecular Devices, LLC, US). Data of the 10 minute continuous recording was acquired at 10 kHz low-pass filtered at 5 kHz. Slices were superfused with standard aCSF that was supplemented with 1 µM Tetrodotoxin (TTX) and 10 µM bicuculline continuously.

### Short-term plasticity

Excitatory postsynaptic currents (EPSCs) were recorded in whole-cell mode using aCSF as described above. Extracellular stimulation (2–3 MΩ resistance) was delivered with a master8 pulse stimulator (A.M.P.I., Israel) at the time intervals: 20, 30, 40, 50, 60, 75, 100, 150, 200, 300 and 400 ms, which was placed in the stratum radiatum of the hippocampus at 80–150 µm distance from the soma. An input/output stimulation response curve was made prior to recording. The 60% of the maximum stimulation intensity was then selected. Series and input resistance were monitored throughout the recordings by means of a voltage step on each sweep. To avoid influence of the overlap between the EPSC response and the next stimulation, kinetic data were obtained from the first stimulation pulse of the 5 and 20 Hz data only (5 *vs.* 20 Hz; all p>0.500). The current amplitudes obtained from

these measurements were not significantly altered (181 ± 12 *vs.* 179 ± 7; unpaired t-test, p=0.866; *Figure 3—figure supplement 1*). Data was analyzed using custom-written Matlab (Mathworks, Natick, MA) scripts. Rise and deactivation times were measured as the 20% to 80% time from baseline to peak amplitude. The desensitization time constant was determined by fitting double exponential curves. The paired pulse ratio (PPR) was calculated as the amplitude ratio of each pulse *vs.* the first pulse: A(n)/A(1st).

## Long term potentiation (LTP)

Excitatory postsynaptic potentials (EPSPs) were recorded in whole-cell current clamp of CA1 pyramidal cells. Slices were superfused at a flow rate of 3–4 mL/minute with standard aCSF with the addition of the GABA_A antagonist bicuculline (10 μM) and kept at 30–31 °C. Extracellular stimulation was delivered on Schaffer collaterals with a theta burst protocol consisting of trains of 5 pulses at 100 Hz, repeated 10 times at 5 Hz. To induce LTP, 3 repetitions of the theta burst were given. Intracellular postsynaptic current injection was performed on the clamped cell to ensure reliable generation of an action potential. Intracellular and extracellular stimulations were paired with a delay of 3 ms (post- minus pre-synaptic stimulation).

Data was discarded if the membrane potential would shift more than 5 mV or the series resistance would change >20%. Series resistance was monitored throughout the recordings by means of a voltage step on each sweep. Recordings were made every 10 s and should last at least 20 minute after LTP induction to be included in the analysis, where only 1 KO slice had recording until 23 minutes after induction. Stimulation input/output tests were made prior to recordings. Input-output relations were plotted as EPSP amplitude *vs.* the stimulus intensity (10–75 μA). The 50% of the maximum stimulation intensity was selected. EPSP amplitude and slope ratios were analyzed as the LTP quantification. Data was binned per minute and averaged for different time intervals. To determine the stability of the baseline, a linear correlation coefficient was estimated ($r^2$), and data was only included if $r^2$ <0.35. EPSP slope was quantified by two methods: fitting to a linear regression the 20% to 80% and by fitting to the initial 2 ms of the rising phase. Slope coefficients of correlation were always >0.95. Slices were included when they displayed a response in at least four stimulus intensities, and intensities were displayed when represented by at least two slices.

## Behavior

### Fear conditioning

All experiments were carried out in a fear conditioning system (TSE Systems; Bad Homburg, Germany). Mice were trained and tested in a Plexiglas chamber (36 × 21 × 20 cm) with a stainless steel grid floor with constant illumination (100–500 lx) and background sound (white noise, 68 dB sound pressure level), which was placed in gray box to shield it from the outside. Before each training or testing session, the chamber was thoroughly cleaned with 70% ethanol.

### Contextual fear conditioning

Training consisted of an exploration period of 180 s, after which a mild 2 s foot shock (0.7 mA) was delivered through the grid floor. Mice were returned to their home cage 30 s after the shock ended. Retrieval tests consisted of re-exposure (180 s) to the training context (conditioned stimulus; CS), at 2 hr or 24 hr after conditioning to determine short-term memory (STM) or long-term memory (LTM), respectively. Freezing was defined as the lack of any movement during 2 s bouts, besides respiration and heartbeat.

### Auditory fear conditioning

In the conditioning box a high-frequency loudspeaker provided a constant auditory background noise (white noise, 68 dB sound pressure level) for 180 s followed by a 30 s 10 kHz tone as conditioned stimulus (70 dB sound pressure level, pulse 5 Hz). The tone was terminated by a 2 s foot shock (0.7 mA). Mice were returned to their home cage 30 s after the shock ended. The auditory memory test was performed 24 hr after conditioning in a novel context (Context B). Mice were placed in the novel context and after 180 s (pre-tone phase) the tone was played for another 180 s. Context B consisted of a similar plexi-glass chamber that was covered with additional visual cues and cleaned with 1% acetic acid before testing. The novel context consisted of a smooth floor

(lacking a grid) in a bright environment (380–480 lx). No background noise was presented during testing. Sample size was based on previous results (*Rao-Ruiz et al., 2011*).

### Approach behavior

Open field, elevated plus maze (EPM) and light/dark box (LDB) tests were performed to study exploratory and anxiety related behavior in wildtype and *Shisa7* KO mice. All behavior was monitored using video tracking (Viewer 2, BIOBSERVE GmbH, Bonn, Germany). Results were analyzed using an unpaired Student's t-test or Mann-Whitney U test if the data did not follow a normal distribution. All data are presented as mean ±SEM.

### Open field

Mice were placed in a corner of a white square open field box (50 × 50 cm, walls 35 cm high, 200 lx). The surface area was divided into nine equally sized squares, and the center square was used as center area. Time spent in the center area and total distance moved were monitored for 10 minutes.

### EPM

Mice were placed on the closed arm of an EPM facing the wall (arms 30 × 6 cm, walls 35 cm high, elevated 50 cm above the ground). The maze was illuminated by a single light bulb (open arms 70 lx, closed arms 30 lx). Visits in open arms, time spent in the open arms and total distance moved were monitored for 5 minutes.

### LDB

Mice were placed into the dark compartment of a light/dark box (25 × 25 cm, 30 cm high, <10 lx) and kept for 60 s until a sliding door provided access to the light compartment (25 × 25 cm, 30 cm high, 625 lx). Time spent in the light compartment and number of visits to the light compartment was tracked during 600 s of free exploration. Only the first 300 s were used for analysis to capture novelty aspects.

## Statistics

Data is presented as mean or median value ±SEM. For HEK293 cell data sets (>20 data points), an outlier analysis was applied (stem-leaf) and extreme values were discarded when they occurred in at least two parameters for the same cell, resulting in omitting 3 cells from GluA1+ Shisa7 for the 1 s glutamate application. For the chemical LTP data, each of the two independent experiments (n = 15 wells per group) was first analyzed on outliers, normality, and statistical differences, after which they were combined. The statistical outliers (WT non-stimulated, DIV14 n = 1, DIV16 n = 3; WT LTP, DIV14 n = 2; *Shisa7* KO non-stimulated, DIV14; *Shisa7* KO LTP, DIV14 n = 1, DIV16 n = 1) corresponded to methodological/biological variation, such as an extraordinary dense culture or overgrowing astrocytes.

For genotype comparisons, Student's t-tests (with or without correction for unequal variation) were applied for normally distributed data (Kolmogorov-Smirnov and Saphiro-Wilk tests) and Mann-Whitney U-tests otherwise. ANOVA tests were carried out for genotype and time (EPSP parameters), or test condition (auditory fear conditioning) as repeated measures. Relative protein quantification, as measured by mass spectrometry, was tested for significance using both a normalized and non-normalized beta-binomial test (*Pham et al., 2010*). Neurite morphology and quantification of spine size and density were tested using multivariate analysis of variance (MANOVA). GluA2 intensity was compared for significance using univariate ANOVA. Statistical significance level was set for p-values<0.05. Only upon reaching significance, post-hoc tests were carried out as single-sided tests. Statistical significance was assessed using Graphpad Prism 5 software (GraphPad Software, La Jolla, CA), and SPSS v20 IBM.

## Acknowledgements

We thank Titia Gebuis, Frank Koopmans, Yvonne Gouwenberg, Joost Hoetjes, Frank den Oudsten, Robert Zalm, Tim Heistek and Jaap Timmerman for their technical assistance. In addition, we thank

Daniel Choquet and Francoise Coussen (IINS, University Bordeaux, France) for their stimulating discussions regarding Shisa7.

## Additional information

### Competing interests
August B Smit: Participates in a holding that owns shares of Sylics BV. The other authors declare that no competing interests exist.

### Funding

| Funder | Grant reference number | Author |
|---|---|---|
| Seventh Framework Programme | HEALTH-2009-2.1.2.1 EU-FP7 SynSys | Marta Ruiperez-Alonso Jasper Stroeder Huib D Mansvelder August B Smit Sabine Spijker |
| Nederlandse Organisatie voor Wetenschappelijk Onderzoek | NWO-ALW #822.02.020 | Remco V Klaassen |
| NBSIK PharmaPhenomics | FES0908 | Leanne J M Schmitz Rolinka J van der Loo August B Smit |
| NBSIK Mouse Phenomics Consortium | BSIK03053 | Priyanka Rao-Ruiz Rolinka J van der Loo August B Smit |
| European Commission | MEST-CT-2005-020919 Neuromics | Priyanka Rao-Ruiz |
| European Commission | MEST-ITN-2008-238686 CerebNet | Jasper Stroeder |
| Nederlandse Organisatie voor Wetenschappelijk Onderzoek | NWO-ALW Vici 865.13.002 | Huib D Mansvelder |
| European Research Council | ERC BrainSignals 281443 | Huib D Mansvelder |
| Nederlandse Organisatie voor Wetenschappelijk Onderzoek | NWO-ALW Vici 016.150.673 / 865.14.002 | Leanne J M Schmitz Sabine Spijker |
| Erasmus Mundus | 159302-1-2009-1-NL-ERA MUNDUS-EMJD | Azra Elia Zamri |

The funders had no role in study design, data collection and interpretation, or the decision to submit the work for publication.

### Author contributions
Leanne J M Schmitz, Conceptualization, Data curation, Formal analysis, Investigation, Visualization, Writing—original draft, Writing—review and editing; Remco V Klaassen, Conceptualization, Data curation, Formal analysis, Investigation, Visualization, Methodology; Marta Ruiperez-Alonso, Conceptualization, Formal analysis, Investigation, Methodology, Writing—original draft; Azra Elia Zamri, Conceptualization, Formal analysis; Jasper Stroeder, Investigation, Methodology; Priyanka Rao-Ruiz, Rolinka J van der Loo, Data curation, Formal analysis, Investigation; Johannes C Lodder, Data curation, Formal analysis, Investigation, Methodology; Huib D Mansvelder, Conceptualization, Supervision, Funding acquisition, Project administration, Writing—review and editing; August B Smit, Conceptualization, Supervision, Funding acquisition, Validation, Writing—original draft, Project administration, Writing—review and editing; Sabine Spijker, Conceptualization, Data curation, Formal analysis, Supervision, Funding acquisition, Visualization, Project administration, Writing—review and editing

### Author ORCIDs

Huib D Mansvelder [iD] http://orcid.org/0000-0003-1365-5340
Sabine Spijker [iD] http://orcid.org/0000-0002-6814-2019

## Ethics

Animal experimentation: All experiments were performed in accordance to Dutch law and licensing agreements using a protocol approved by the Animal Ethics Committee of the VU University Amsterdam.

## Decision letter and Author response

Decision letter https://doi.org/10.7554/eLife.24192.029
Author response https://doi.org/10.7554/eLife.24192.030

---

## Additional files

### Supplementary files

• Supplementary file 1. The functional contribution of Shisa family members (Shisa9, Shisa6, Shisa7) and TARP γ−8 on AMPAR complexes are listed for the respective in vitro, ex/in vivo methods used. In vitro data on TARP γ−8 are from *Milstein et al. (2007)*. The Shisa9 and TARP γ−8 ex/in vivo data originate from *Khodosevich et al. (2014)*, in which experiments were performed using WT, KO and overexpression conditions in the DG, which is regionally more appropriate for Shisa9 than overexpression in CA1 (*von Engelhardt et al., 2010*), and from the CA1 for *TARP γ−8* KO (*Rouach et al., 2005*). The Shisa6 ex/in vivo data stem from WT *vs.* KO comparisons in CA1 pyramidal cells activated by (electrically-evoked) CA3 input (Schaffer collaterals (*Klaassen et al., 2016*)), similar as used for Shisa7 (bold). NA: Not applicable, not measured.
DOI: https://doi.org/10.7554/eLife.24192.026

• Transparent reporting form
DOI: https://doi.org/10.7554/eLife.24192.027

---

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
