## [Decision Letter]

Thank you for submitting your article "The AMPA receptor-associated protein Shisa7 regulates hippocampal synaptic function and contextual memory" for consideration by *eLife*. Your article has been reviewed by three peer reviewers, one of whom is a member of our Board of Reviewing Editors, and the evaluation has been overseen by Gary Westbrook as the Senior Editor. The reviewers have opted to remain anonymous. The reviewers have discussed the reviews with one another and the Reviewing Editor has drafted this decision to help you prepare a revised submission.

Summary:

Auxiliary subunits of AMPARs play critical roles in regulating AMPAR activity in vivo, and CKAMP/Shisa family proteins have been shown to function as AMPAR auxiliary subunits. Here Schmitz et al. show that *Shisa7/CKAMP59* is enriched in the mouse PSD fraction. Overexpressed Flag-Shisa7 in *Shisa7* KO primary cultured neurons co-localize with Homer1 and co-IPed with AMPARs. Furthermore, Shisa7 modulates recovery from desensitization of GluA1 expressed in HEK cells without affecting the deactivation and desensitization. In the Shisa7 KO mice the decay kinetics of mEPSC and eEPSC were significantly faster without obvious changes in expressions and localizations of AMPARs and other synaptic proteins. Furthermore, LTP and contextual fear memory were impaired in *Shisa7* KO mice.

Even though the interaction of Shisa7 and AMPA receptor subunits was previously reported, this study advances our understanding of AMPA receptor auxiliary subunits and of Shisa proteins in particular. The results show an unexpected role of Shisa7 in regulating the recovery from desensitization. The findings that induction and LTP is impaired in absence of Shisa7 and that Shisa7 plays region-specific roles in memory processes are of additional relevance. Further, the experiments are well performed. Important open questions exist, though, about the mechanisms of Shisa7 function. In addition, the manuscript contains a number of imprecise descriptions and lack of explanations, which require careful attention in the revised manuscript.

Essential revisions:

1) The mechanism by which Shisa7 impacts LTP is unclear. Is the desensitization rate of AMPA receptors causally linked to LTP induction or maintenance? If there is no such evidence, the authors should test alternative explanations. Even though Shisa7 does not impact synaptic AMPA receptor abundance at steady state, Shisa7 may regulate the activity-dependent trafficking of AMPA receptors to synapses as LTP is induced and their synaptic retention during LTP maintenance. This can be addressed e.g. using quantitative surface staining after chemical LTP in cultured neurons.

2) To conclude specific changes in AMPARs, analysis of NMDAR-mediated EPSCs is necessary. A ratio of AMPAR/NMDAR-mediated evoked EPSC amplitude and NMDAR decay kinetics in Shisa7 KO mice should be provided.

3) Shisa7 prolongs AMPAR currents and regulates LTP. However, the data seem to provide little insights into the underlying mechanism(s). The authors should try virus-mediated rescue experiments and demonstrate that the observed changes in mEPSC currents and synaptic plasticity can be rescued by wild-type Shisa7 but not by a mutant Shisa7, which e.g. lacks the PSD-95-binding C-terminus.

[Editors' note: further revisions were requested prior to acceptance, as described below.]

Thank you for resubmitting your work entitled "The AMPA receptor-associated protein Shisa7 regulates hippocampal synaptic function and contextual memory" for further consideration at *eLife*. Your revised article has been favorably evaluated by Gary Westbrook (Senior editor) and a Reviewing editor.

The manuscript has been improved but there are some remaining issues, specifically regarding essential point 3 in the prior decision letter that we would like you to address before acceptance. That point asked for a rescue experiment by wildtype Shisa7, but not by mutant Shisa7 that for example lacked the PSD-binding C terminus. In your rebuttal you indicate that you did not think these experiments would be informative as "…we observed previously that overexpression of Shisa7 results in a loss of spines in WT and KO culture alike. Apparently, the acute increase in Shisa7 levels has a major impact on synapse integrity. For this same reason, it would not make sense to include a quantitative analysis of the Shisa7 – PSD-95 colocalization (Figure 1C). Rather, this image was shown for qualitative purposes, namely that the majority of Shisa7 is in the post-synaptic density together with PSD-95, as our biochemical data shows. Thus, given the fact that the viral expression level of Shisa7 is hard to control and leads to spine loss, we think a rescue experiment will not lead to justified conclusions…".

We accept your decision not to pursue these experiments in the manuscript. However, we think readers might have similar questions. Thus a discussion of these issues and your observations with overexpression in WT and KO cultures deserves discussion in the text of your manuscript. Thus please revise the discussion in that regard.

---

## [Author Response]

Essential revisions:1) The mechanism by which Shisa7 impacts LTP is unclear. Is the desensitization rate of AMPA receptors causally linked to LTP induction or maintenance? If there is no such evidence, the authors should test alternative explanations. Even though Shisa7 does not impact synaptic AMPA receptor abundance at steady state, Shisa7 may regulate the activity-dependent trafficking of AMPA receptors to synapses as LTP is induced and their synaptic retention during LTP maintenance. This can be addressed e.g. using quantitative surface staining after chemical LTP in cultured neurons.

We executed the experiment suggested by the reviewers. We have performed chemical LTP experiments, quantified the AMPAR surface staining in cultured neurons for WT and KO. This data is in agreement with the LTP data as obtained in slices, namely that AMPAR recruitment to the membrane is hampered in the *Shisa7* KO. This data is now added to Figure 6 and its supplemental figure.

2) To conclude specific changes in AMPARs, analysis of NMDAR-mediated EPSCs is necessary. A ratio of AMPAR/NMDAR-mediated evoked EPSC amplitude and NMDAR decay kinetics in Shisa7 KO mice should be provided.

We have now measured the amplitude of NMDAR-mediated EPSCs in WT and *Shisa7* KO slices. There is no difference in this; the data is provided in the supplement of Figure 5 (input-output), next to that of the AMPAR.

3) Shisa7 prolongs AMPAR currents and regulates LTP. However, the data seem to provide little insights into the underlying mechanism(s). The authors should try virus-mediated rescue experiments and demonstrate that the observed changes in mEPSC currents and synaptic plasticity can be rescued by wild-type Shisa7 but not by a mutant Shisa7, which e.g. lacks the PSD-95-binding C-terminus.

We respectfully disagree that rescue of *Shisa7 expression* will give insight into the *underlying mechanism* of disrupted LTP in the *Shisa7* KO.

In particular, we have stayed away from these experiments, as we observed previously that overexpression of Shisa7 results in a loss of spines in WT and KO culture alike. Apparently, the acute increase in Shisa7 levels has a major impact on synapse integrity. For this same reason, it would not make sense to include a quantitative analysis of the Shisa7 – PSD-95 colocalization (Figure 1C). Rather, this image was shown for qualitative purposes, namely that the majority of Shisa7 is in the post-synaptic density together with PSD-95, as our biochemical data shows.

Thus, given the fact that the viral expression level of Shisa7 is hard to control and leads to spine loss, we think a rescue experiment will not lead to justified conclusions. Therefore, we have refrained from doing these experiments, and have rather focussed on acquiring relevant data for essential points 1 and 2.

[Editors' note: further revisions were requested prior to acceptance, as described below.]

The manuscript has been improved but there are some remaining issues, specifically regarding essential point 3 in the prior decision letter that we would like you to address before acceptance. That point asked for a rescue experiment by wildtype Shisa7, but not by mutant Shisa7 that for example lacked the PSD-binding C terminus. In your rebuttal you indicate that you did not think these experiments would be informative as "…we observed previously that overexpression of Shisa7 results in a loss of spines in WT and KO culture alike. Apparently, the acute increase in Shisa7 levels has a major impact on synapse integrity. For this same reason, it would not make sense to include a quantitative analysis of the Shisa7 – PSD-95 colocalization (Figure 1C). Rather, this image was shown for qualitative purposes, namely that the majority of Shisa7 is in the post-synaptic density together with PSD-95, as our biochemical data shows. Thus, given the fact that the viral expression level of Shisa7 is hard to control and leads to spine loss, we think a rescue experiment will not lead to justified conclusions…".We accept your decision not to pursue these experiments in the manuscript. However, we think readers might have similar questions. Thus a discussion of these issues and your observations with overexpression in WT and KO cultures deserves discussion in the text of your manuscript. Thus please revise the discussion in that regard.

On your suggestion, we have added the observation of spine loss upon overexpression of Shisa7 (Results section, paragraph one), and added in the Discussion a sentence on absence of mobility experiments and the inability to perform rescue experiments where it concerned the PDZ ligand motif (subsection “Shisa7 is required for synaptic plasticity-dependent changes in AMPAR expression”).

Furthermore, the supplemental methods have been incorporated into the normal Materials and methods section, and the table giving an overview of effects of Shisa proteins and TARP g8 is now Supplementary file 1. I think with that we fulfilled all questions raised so far.